# Are transient protein-protein interactions more dispensable?

**Mohamed Ali Ghadie**, **Yu Xia**\*

Department of Bioengineering, McGill University, Montreal, Canada

\* brandon.xia@mcgill.ca

## Abstract

Protein-protein interactions (PPIs) are key drivers of cell function and evolution. While it is widely assumed that most permanent PPIs are important for cellular function, it remains unclear whether transient PPIs are equally important. Here, we estimate and compare dispensable content among transient PPIs and permanent PPIs in human. Starting with a human reference interactome mapped by experiments, we construct a human structural interactome by building three-dimensional structural models for PPIs, and then distinguish transient PPIs from permanent PPIs using several structural and biophysical properties. We map common mutations from healthy individuals and disease-causing mutations onto the structural interactome, and perform structure-based calculations of the probabilities for common mutations (assumed to be neutral) and disease mutations (assumed to be mildly deleterious) to disrupt transient PPIs and permanent PPIs. Using Bayes' theorem we estimate that a similarly small fraction (<~20%) of both transient and permanent PPIs are completely dispensable, i.e., effectively neutral upon disruption. Hence, transient and permanent interactions are subject to similarly strong selective constraints in the human interactome.

## Author summary

All cellular functions are driven by interactions between different biomolecules in the cell. Among these interactions are protein-protein interactions, abbreviated as PPIs, in which two proteins physically bind to each other to perform a specific molecular function. PPIs are often divided into two categories: transient PPIs where two proteins bind to each other only for a short time and then break apart, and permanent PPIs where two proteins bind to each other permanently thus forming a permanent protein complex. Considering their permanent nature, it is generally assumed that permanent PPIs are important for cellular function, whereas it remains unclear whether transient PPIs are equally important. Here, we estimate the fractions of transient PPIs and permanent PPIs in human that can be removed from the cell without harming human fitness. We do this by constructing three-dimensional structural models for PPIs which allow us to predict the probabilities for disease-causing mutations and non-disease mutations from healthy individuals to disrupt transient PPIs and permanent PPIs. Using these probabilities, we estimate that similar to permanent PPIs only a small fraction of transient PPIs (<~20%) can be disrupted

**Data Availability Statement:** The source data underlying all figures are included in this article and its supplementary data files. Code that was used to build the human structural interactome is available at the following GitHub repository: https://github.com/MohamedGhadie/build_structural_interactome. Code that was used to map mutations

onto the human structural interactome and calculate dispensable content among transient PPIs and permanent PPIs is available at the following GitHub repository: https://github.com/MohamedGhadie/dispensable_transient_ppis.

**Funding:** This work was supported by Natural Sciences and Engineering Research Council of Canada grants RGPIN-2019-05952 and RGPAS-2019-00012, Canada Foundation for Innovation grant JELF-33732, and Canada Research Chairs program to Y.X.. The funders had no role in study design, data collection and analysis, decision to publish, or preparation of the manuscript.

**Competing interests:** The authors have declared that no competing interests exist.

without harming fitness, indicating that most transient and permanent PPIs are important for cellular function in human.

## Introduction

Protein-protein interactions (PPIs) implement thousands of functions at the molecular level, taking part in almost every biological process in the cell. Hence, the collective network of PPIs, commonly known as the interactome network, has been imperative for our understanding of cell function [1,2], disease [3–7], and evolution [8–11], especially when combined with protein structural information [12–21]. Nonetheless, PPIs across the interactome network are very diverse in their structural, biophysical and spatiotemporal properties [22–24]. Based on their binding patterns across time and space, PPIs are divided into two categories: transient PPIs and permanent PPIs [22,23]. These binding patterns are largely controlled by the strength of interaction as well as co-expression of interaction partners [25–27]. A PPI is transient in time if the two interaction partners form a weak interaction only for a short period of time and then break apart [22,23,27]. On the other hand, a PPI is permanent in time if the two interaction partners form a strong interaction that continues to exist without breaking apart thus forming a permanent protein complex [22,27]. PPIs can also be transient or permanent in space [28]. A PPI is permanent in space if the two interaction partners always co-express to form stoichiometric complexes in the same tissues or cell types, and transient otherwise [28,29]. While systems biology studies typically rely on the assumption that most PPIs in human are important for cellular function [2,5,30,31], we recently estimated that indeed only <~20% of the human interactome is completely dispensable, i.e., effectively neutral upon disruption by mutation [20]. Completely dispensable PPIs are those PPIs that are disrupted in the presence of a mutation at the binding interface, i.e., they are completely eliminated from the interactome as a result of mutation, however their elimination from the interactome has no measurable deleterious impact on organismal fitness [20]. These completely dispensable PPIs are different than other PPIs that are robust to the presence of mutations at the interface hence are not eliminated from the interactome as a result of mutation [9,11,32]. However, our estimate of dispensable content in the human interactome represents an average over the entire interactome. It remains an open question whether transient PPIs have more dispensable content than other permanent PPIs [33–35].

The question of how important transient PPIs are to cellular function is crucial to our understanding of cell systems biology and human disease [5,30,31]. In the absence of any quantitative model for measuring the importance of transient PPIs, our judgement relies heavily on different studies leading to diverging conclusions. On the one hand, many transient PPIs have been found to play important roles in defining the structure of interactome networks, such as regulating interactome modularity [26,27] and guiding the formation of obligate protein complexes [36,37]. Other transient PPIs are known to participate in multiple cellular pathways and biochemical processes, including secretory pathways [38], signal transduction [39–41], immune response [42], chaperone-guided protein folding [43,44], apoptosis [45], and tumor suppression [46]. While these studies show that many transient PPIs play important roles in cellular function and human disease, it remains unclear whether this observed functional significance generalizes to most transient PPIs. On the other hand, transient PPIs differ from permanent PPIs in their structural composition and binding dynamics [23,27,47]. They tend to occur among certain protein hubs, known as "date hubs", which interact with multiple partners in a mutually exclusive manner using the same binding interface

[12,26]. This behaviour is contrary to that of permanent PPIs which tend to occur among a different type of protein hubs, known as "party hubs", which interact with multiple partners simultaneously using multiple binding interfaces [12,26]. Mutually exclusive transient PPIs are often mediated through short linear motifs that typically occur in intrinsically disordered regions [27,48,49]. These binding motifs tend to be smaller in surface area and contain less hydrophobic residues than interfaces of permanent PPIs, thus they bind with weaker affinities [27]. Linear motifs also evolve very rapidly [27,49], contributing in part to the higher rate of rewiring among transient PPIs compared to permanent PPIs [35,50–52]. Indeed, empirical studies in phospho-proteomics and molecular evolution estimate that as much as ~65% of transient phosphorylation sites in yeast are unconstrained under evolution and may have no important function [33,34].

Detecting transient PPIs in experiments and through computational predictions is very challenging [23,47,53]. With different limitations and biases associated with experimental techniques for detecting PPIs [54,55], large-scale datasets of transient PPIs are currently not available. However, databases that curate PPI sequence and structural information such as linear motifs [56,57], domain-motif interactions [58], domain-domain interactions [59], three-dimensional protein structures [60], and binding affinities [61] from both experimental and computational studies can be used to predict transient and permanent PPIs. Some studies have used PPI binding affinity measurements from experiments to identify transient and permanent interactions [61,62]. The scope of these studies is limited by the small number of PPIs with affinity data available from experiments. Computational studies have used interface structural information [12,63], protein sequence information [64] and machine learning models [65,66] to predict transient and permanent PPIs. Other studies made use of gene expression data which does not rely on experimentally-solved protein structures [25,67]. Since the criteria and information that were used to detect transient interactions vary among these studies, each study has its own biases and sources of error. Moreover, with the large increase in PPI datasets recently mapped by experiments in human [68,69], and given the difficulty in identifying transient PPIs in experiments, there is a great need for new computational efforts to accurately classify transient and permanent interactions in these new datasets, taking into account different structural and biophysical properties that distinguish transient PPIs from permanent PPIs.

Here, in an effort to answer the long-standing question of the overall importance of transient PPIs, we provide a quantitative measure of their importance and compare them with permanent PPIs by estimating dispensable content among both types of PPIs, using the same procedure we developed before to estimate dispensable content in the overall human interactome [11,20]. Starting with a high-quality human reference interactome mapped by experiments, we apply homology modelling similar to [20] to construct a high-resolution three-dimensional (3D) human structural interactome with PPI binding interfaces annotated at the residue level (Fig 1A). This structural interactome is much larger than our previously constructed structural interactome in [20]. Next, we label each PPI in the structural interactome as either transient or permanent based on different structural, biophysical and spatiotemporal properties (Fig 1A). We map common mutations from healthy individuals as well as Mendelian disease-causing mutations onto the human structural interactome and perform structure-based calculations similar to [20] to predict the edgotype [70] for each mutation, i.e., the precise pattern of PPI perturbations as a result of each mutation. Unlike in [20], here the edgotype of a mutation is defined separately for each group of PPIs, transient and permanent. Finally, we integrate our edgotype predictions into the same Bayesian framework we used in [20] and calculate the fraction of PPIs that are completely dispensable, i.e., effectively neutral upon disruption by mutation, among each group of PPIs (Fig 1B). Overall, we estimate that only $<~20\%$ of transient PPIs are completely dispensable, and the remaining are at least mildly

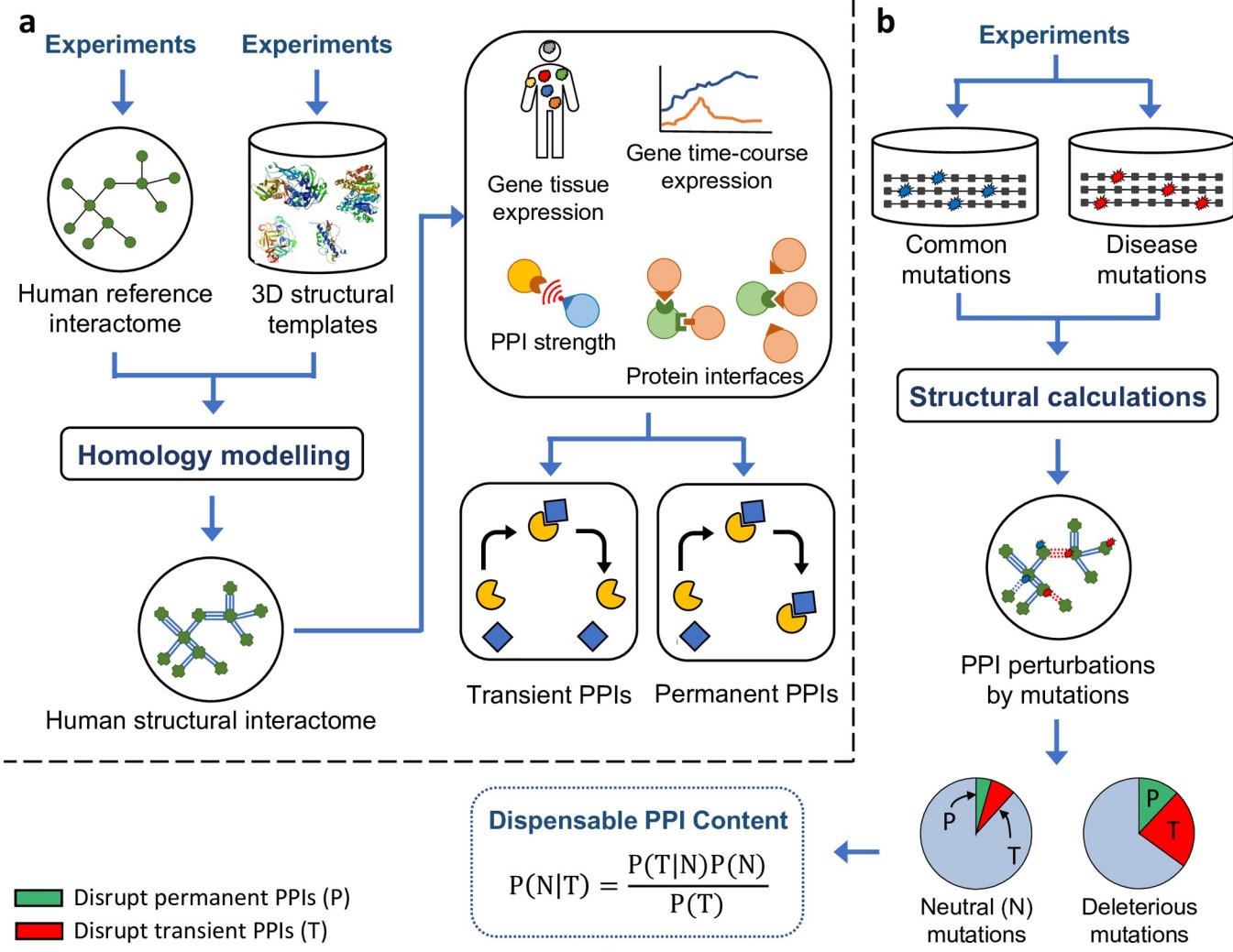

**Fig 1. Computational pipeline for structure-based calculation of dispensable PPI content.** (a) Construction of the human structural interactome and subsequent classification of transient PPIs and permanent PPIs using structural, biophysical and spatiotemporal properties. (b) Structure-based prediction of PPI perturbations by missense mutations, and subsequent calculation of dispensable PPI content.

deleterious upon disruption. We also estimate that $<\sim 20\%$ of permanent PPIs are completely dispensable. These two estimates are comparable to our estimate of dispensable content calculated over all PPIs in the interactome network, using both predicted mutation edgotypes as well as mutation edgotypes obtained from experiments [4]. Our results reveal that, similar to permanent PPIs, most transient PPIs ($>\sim 80\%$) are important to cellular function hence deleterious upon disruption. Our results also suggest that transient and permanent interactions are subject to similarly strong selective constraints in the human interactome.

## Results

### Structure-based prediction of mutation edgotypes

We obtained two high-quality human reference interactomes that were mapped by experiments: the HuRI interactome consisting of PPIs identified most recently in yeast two-hybrid (Y2H) screens [68], and the literature-curated interactome consisting of PPIs reported by at

least two independent experiments in the IntAct database [71]. From each reference interactome, we constructed a human structural interactome by building 3D structural models for PPIs using homology modelling based on experimentally determined structural templates in PDB [60] (Fig 1A). As a result, we obtained two high-resolution human structural interactomes with PPI binding interfaces annotated at the residue level: the Y2H structural interactome (Y2H-SI) consisting of 1,916 PPIs among 1,468 proteins (S1A Data), and the literature-derived structural interactome (Lit-SI) consisting of 4,676 PPIs among 3,445 proteins (S1B Data). These two structural interactomes are much larger than the structural interactomes we had previously constructed in [20]. Y2H-SI which was derived in this study from the recent HuRI dataset is 4 times larger in number of PPIs than the Y2H structural interactome in [20], which was derived from the much smaller HI-II-14 dataset [72]. In addition, Lit-SI in this study is also 1.4 times larger in number of PPIs than the literature-derived structural interactome in [20].

Next, we mapped Mendelian disease-causing missense mutations from ClinVar [73] and common missense mutations not associated with any disease from dbSNP [74] onto the two human structural interactomes, Y2H-SI and Lit-SI (Fig 1B). As a result, we obtained 348 disease mutations and 1,080 common mutations in Y2H-SI (S2A and S2B Data), as well as 1,572 disease mutations and 2,867 common mutations in Lit-SI (S2C and S2D Data). These mutations cover ~39% of proteins in Y2H-SI and ~42% of proteins in Lit-SI, thus spanning a significant part of the human structural interactome. In addition, these results represent a 2.74-fold and 1.34-fold increase in the number of mutations that were successfully mapped onto Y2H-SI and Lit-SI, respectively, compared to our previous results in [20].

Using our structural interactomes, we performed structure-based calculations to predict the edgotype [70] for each mutation, i.e., the precise pattern of interactome perturbations created by each mutation (Fig 1B). The edgotype of a mutation is edgetic if it disrupts PPIs by disrupting a specific binding interface, quasi-null if it disrupts all PPIs by disrupting overall protein stability, or quasi-wild-type if it does not disrupt any PPI [4]. We predict that at a mutation edgetically disrupts a PPI if and only if the mutation occurs on the interface mediating that PPI and causes a change in PPI binding free energy ($\Delta\Delta G$) larger than 0.5 kcal/mol (S2 Data), as calculated on the PPI structural model using FoldX [75] (S3 Data).

## Dispensable content among weak transient PPIs and strong permanent PPIs

One biophysical property that distinguishes transient PPIs from permanent PPIs is the strength of interaction. While transient PPIs tend to form weak molecular interactions that are easily broken apart, permanent PPIs tend to form stronger interactions that are harder to break [27]. Thus, we estimated dispensable content among weak transient PPIs as well as strong permanent PPIs in both structural interactomes Y2H-SI and Lit-SI. We consider a PPI to be weak if it has a binding free energy $\Delta G \geq$ -25 kcal/mol as calculated by FoldX on the PPI structural template, otherwise we consider the PPI to be strong (S4 Data). According to this definition, 57% of PPIs in Y2H-SI and 66% of PPIs in Lit-SI are considered to be weak interactions (S5 Data).

We first calculated dispensable content among weak transient PPIs and strong permanent PPIs in Y2H-SI, using the Bayesian framework we had previously developed [11,20] and describe here in the Methods section. We assume that mutations are either effectively neutral (similar to synonymous mutations), mildly deleterious, or strongly detrimental (similar to nonsense mutations that introduce premature stop codons). In addition, we assume that common mutations from healthy individuals are effectively neutral, that Mendelian disease-

**Table 1. Mutation edgotype data obtained from predictions.** Number of common mutations and disease mutations that edgetically disrupt transient and permanent PPIs defined by different structural, biophysical and spatiotemporal properties in the two human structural interactomes Y2H-SI and Lit-SI.

| PPI properties | | | | Common mutations | | | Disease mutations | | |
|---|---|---|---|---|---|---|---|---|---|
| Transient PPIs | Permanent PPIs | Expression data | SI | Total | Disrupt transient PPIs | Disrupt permanent PPIs | Total | Disrupt transient PPIs | Disrupt permanent PPIs |
| Weak | Strong | | Y2H | 1,080 | 3 | 13 | 348 | 7 | 36 |
| | | | Lit | 2,867 | 27 | 29 | 1,572 | 51 | 68 |
| Transient in time | Permanent in time | Time-course (GEO) | Y2H | 1,080 | 7 | 9 | 347 | 18 | 24 |
| | | | Lit | 2,867 | 16 | 40 | 1,571 | 57 | 61 |
| Transient in space | Permanent in space | Tissue (Illumina) | Y2H | 1,075 | 4 | 7 | 336 | 12 | 19 |
| | | | Lit | 2,859 | 19 | 29 | 1,563 | 61 | 49 |
| | | Tissue (Fantom5) | Y2H | 1,080 | 3 | 13 | 345 | 23 | 17 |
| | | | Lit | 2,862 | 19 | 32 | 1,570 | 57 | 60 |
| Unbalanced over time | Balanced over time | Time-course (GEO) | Y2H | 1,080 | 1 | 15 | 347 | 8 | 34 |
| | | | Lit | 2,867 | 12 | 44 | 1,570 | 32 | 85 |
| Unbalanced over space | Balanced over space | Tissue (Illumina) | Y2H | 1,080 | 11 | 5 | 343 | 21 | 17 |
| | | | Lit | 2,866 | 32 | 23 | 1,567 | 58 | 56 |
| | | Tissue (Fantom5) | Y2H | 1,080 | 6 | 10 | 344 | 22 | 17 |
| | | | Lit | 2,862 | 22 | 29 | 1,568 | 48 | 67 |
| 1–4 mutually exclusives | No mutually exclusives | | Y2H | 1,079 | 4 | 2 | 348 | 24 | 8 |
| | | | Lit | 2,865 | 30 | 10 | 1,566 | 50 | 34 |
| ≥5 mutually exclusives | | | Y2H | 1,079 | 9 | | 348 | 11 | |
| | | | Lit | 2,865 | 14 | | 1,566 | 29 | |

causing mutations are mildly deleterious on average, and that strongly detrimental mutations are predominantly quasi-null (i.e., disrupt overall protein stability) rather than edgetic. Here, it is important to make a clear distinction between a mutation's fitness effect on the one hand and its impact on the interactome network on the other hand. Both effectively neutral mutations and deleterious mutations may or may not disrupt PPIs, hence they both may or may not be edgetic.

Therefore, from our edgotype predictions in Y2H-SI shown in Table 1, we obtained the probabilities for effectively neutral mutations (N) to edgetically disrupt weak transient PPIs (T): $P(T|N) = 3/1080 = 0.3\%$, and to edgetically disrupt strong permanent PPIs (P): $P(P|N) = 13/1080 = 1.2\%$ (Fig 2A and Table 2). We also obtained the probabilities for mildly deleterious mutations (M) to edgetically disrupt weak transient PPIs (T): $P(T|M) = 7/348 = 2\%$, and to edgetically disrupt strong permanent PPIs (P): $P(P|M) = 36/348 = 10.3\%$ (Fig 2A and Table 2). From these probabilities, we found that the propensity for neutral mutations to disrupt weak transient PPIs relative to mildly deleterious mutations is as low as that among strong permanent PPIs (0.15 for transient PPIs, and 0.12 for permanent PPIs) (Table 2). This similar low enrichment of PPI disruptions by neutral mutations among transient and permanent PPIs already suggests that both types of PPIs are as likely to be effectively neutral upon disruption by mutation.

Next, we obtained from Kryukov et al. [76] the probabilities for new missense mutations to be effectively neutral (N), mildly deleterious (M), or strongly detrimental (S): $P(N) = 27\%$, $P(M) = 53\%$, $P(S) = 20\%$. We then integrated these numbers into Eq 1 in the Methods section to calculate the probability for a new missense mutation to edgetically disrupt a weak transient PPI (T):

$$P(T) = P(T|N)P(N) + P(T|M)P(M) + P(T|S)P(S) = 1.1\%$$

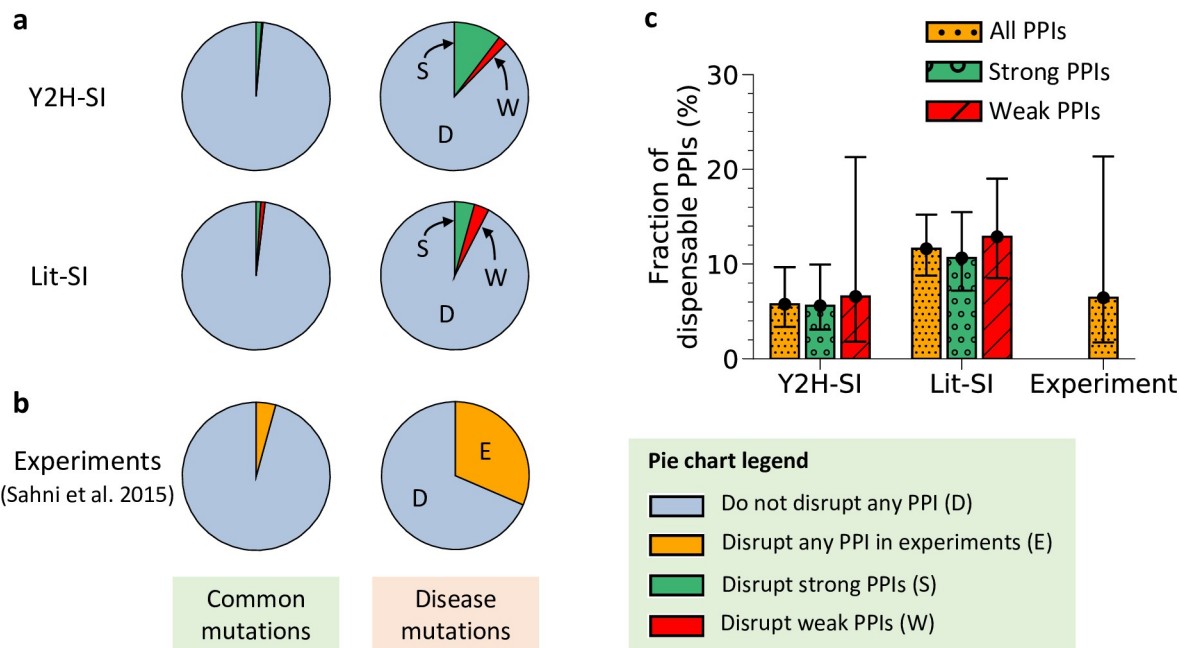

**Fig 2. Dispensable content among weak transient PPIs and strong permanent PPIs.** (a) Fractions of common mutations and disease mutations that edgetically disrupt weak transient PPIs and strong permanent PPIs in the two human structural interactomes Y2H-SI and Lit-SI. (b) Fractions of common mutations and disease mutations that edgetically disrupt any PPI in the human interactome as determined by experiments of Sahni et al. [4]. (c) Dispensable content among weak transient PPIs, strong permanent PPIs, and all PPIs in human calculated using predicted mutation edgotypes in the two human structural interactomes Y2H-SI and Lit-SI, as well as using mutation edgotypes determined by experiments of Sahni et al. [4]. Error bars represent 95% confidence intervals.

where P(T|S) is assumed to be approximately 0. Finally, using Bayes' theorem (Eq 2 in Methods), we calculated the probability for a missense mutation that edgetically disrupts a weak transient PPI (T) to be effectively neutral (N):

$$P(N|T) = \frac{P(T|N)P(N)}{P(T)} = 6.6\%$$

Therefore, since most mutations that edgetically disrupt weak transient PPIs in Y2H-SI (60%) disrupt only one PPI, we estimated that ~7% of weak transient PPIs in human are completely dispensable, i.e., effectively neutral upon disruption, with a 95% confidence interval of ~2–21% (Fig 2C and Table 2).

We applied the same procedure again to calculate dispensable content among strong permanent PPIs in Y2H-SI. From our edgotype predictions shown in Table 1, we obtained the probability for neutral mutations (N) to edgetically disrupt strong permanent PPIs (P): P(P|N) = 1.2%, and the probability for mildly deleterious mutations (M) to edgetically disrupt strong permanent PPIs (P): P(P|M) = 10.3% (Fig 2A and Table 2). We then integrated these edgotype probabilities into Eq 1 in the Methods section to calculate the probability for a new missense mutation to edgetically disrupt a strong permanent PPI (P):

$$P(P) = P(P|N)P(N) + P(P|M)P(M) + P(P|S)P(S) = 5.8\%$$

where P(P|S) is assumed to be approximately 0. Using Bayes' theorem again (Eq 2 in Methods), we calculated the probability for a missense mutation that edgetically disrupts a strong

**Table 2. Dispensable content among transient and permanent PPIs.**

| PPI properties | | | | Transient PPIs | | | | Permanent PPIs | | | |
|---|---|---|---|---|---|---|---|---|---|---|---|
| Transient PPIs | Permanent PPIs | Expression data | SI | P(T\|N) | P(T\|M) | $\frac{P(T\|N)}{P(T\|M)}$ | P(N\|T) | P(P\|N) | P(P\|M) | $\frac{P(P\|N)}{P(P\|M)}$ | P(N\|P) |
| Weak | Strong | | Y2H | 0.3 | 2.0 | 0.15 | 6.6 | 1.2 | 10.3 | 0.12 | 5.6 |
| | | | Lit | 0.9 | 3.2 | 0.28 | 12.9 | 1.0 | 4.3 | 0.23 | 10.6 |
| Transient in time | Permanent in time | Time-course (GEO) | Y2H | 0.6 | 5.2 | 0.12 | 6 | 0.8 | 6.9 | 0.12 | 5.8 |
| | | | Lit | 0.6 | 3.6 | 0.17 | 7.3 | 1.4 | 3.9 | 0.36 | 15.5 |
| Transient in space | Permanent in space | Tissue (Illumina) | Y2H | 0.4 | 3.6 | 0.11 | 5 | 0.7 | 5.7 | 0.12 | 5.5 |
| | | | Lit | 0.7 | 3.9 | 0.18 | 8 | 1.0 | 3.1 | 0.32 | 14.2 |
| | | Tissue (Fantom5) | Y2H | 0.3 | 6.7 | 0.04 | 2.1 | 1.2 | 4.9 | 0.24 | 11.1 |
| | | | Lit | 0.7 | 3.6 | 0.19 | 8.5 | 1.1 | 3.8 | 0.29 | 13 |
| Unbalanced over time | Balanced over time | Time-course (GEO) | Y2H | 0.1 | 2.3 | 0.04 | 2 | 1.4 | 9.8 | 0.14 | 6.7 |
| | | | Lit | 0.4 | 2.0 | 0.20 | 9.5 | 1.5 | 5.4 | 0.28 | 12.6 |
| Unbalanced over space | Balanced over space | Tissue (Illumina) | Y2H | 1.0 | 6.1 | 0.16 | 7.8 | 0.5 | 5.0 | 0.10 | 4.5 |
| | | | Lit | 1.1 | 3.7 | 0.30 | 13.3 | 0.8 | 3.6 | 0.22 | 10.3 |
| | | Tissue (Fantom5) | Y2H | 0.6 | 6.4 | 0.09 | 4.2 | 0.9 | 4.9 | 0.18 | 8.7 |
| | | | Lit | 0.8 | 3.1 | 0.26 | 11.3 | 1.0 | 4.3 | 0.23 | 10.8 |
| 1–4 mutually exclusives | No mutually exclusives | | Y2H | 0.4 | 6.9 | 0.06 | 2.7 | 0.2 | 2.3 | 0.09 | 4.0 |
| | | | Lit | 1.0 | 3.2 | 0.31 | 14.3 | 0.3 | 2.2 | 0.14 | 7.6 |
| ≥5 mutually exclusives | | | Y2H | 0.8 | 3.2 | 0.25 | 11.9 | | | | |
| | | | Lit | 0.5 | 1.9 | 0.26 | 11.9 | | | | |

Edgotype probabilities for neutral and mildly deleterious mutations calculated directly from edgotype numbers in Table 1, assuming that common mutations are effectively neutral (N) and that disease mutations are mildly deleterious (M) on average. The resulting dispensable contents P(N|T) and P(N|P) among both transient (T) and permanent (P) PPIs were calculated using Bayes' theorem. Columns represent the following, SI: structural interactome, P(T|N): probability (%) for neutral mutations (N) to edgetically disrupt transient PPIs (T), P(T|M): probability (%) for mildly deleterious mutations (M) to edgetically disrupt transient PPIs (T), P(N|T): dispensable content among transient PPIs defined as the probability (%) for transient PPIs to be effectively neutral upon disruption, P(P|N): probability (%) for neutral mutations (N) to edgetically disrupt permanent PPIs (P), P(P|M): probability (%) for mildly deleterious mutations (M) to edgetically disrupt permanent PPIs (P), P(N|P): dispensable content among permanent PPIs defined as the probability (%) for permanent PPIs to be effectively neutral upon disruption.

permanent PPI (P) to be effectively neutral (N):

$$P(N|P) = \frac{P(P|N)P(N)}{P(P)} = 5.6\%$$

Thus, we estimated that ~6% of strong permanent PPIs in human are completely dispensable with a 95% confidence interval of ~3–10% (Fig 2C and Table 2).

Finally, we repeated the same calculations using edgotype predictions in Lit-SI shown in Table 1. Similar to Y2H-SI, we found a similar enrichment of PPI disruptions by neutral mutations compared to mildly deleterious mutations among both weak transient PPIs and strong permanent PPIs (Fig 2A and Table 2). Thus, we estimated that ~13% of weak transient PPIs in Lit-SI are completely dispensable with a 95% confidence interval of ~9–19% (Fig 2C and Table 2), and that ~11% of strong permanent PPIs in Lit-SI are completely dispensable with a 95% confidence interval of ~7–15% (Fig 2C and Table 2). Overall, our calculations reveal that dispensable content among both types of PPIs in human, transient and permanent, is below ~20%. This estimate is comparable to our estimates of dispensable content calculated among all PPIs together using predicted mutation edgotypes in Y2H-SI and Lit-SI as well as mutation edgotypes obtained from experiments [4] (Fig 2C).

## Dispensable content among temporally transient PPIs

A second property that distinguishes transient PPIs from permanent PPIs is the time duration of interaction. A PPI is transient in time if the interaction partners bind to each other only for a short period of time and then break apart. On the other hand, a PPI is permanent in time if the interaction partners bind to each other without breaking apart thus forming a permanent protein complex [27] (Fig 3A). Date hubs in particular bind to different partners at different points in time through short-term transient interactions, whereas party hubs bind to different partners at the same time through permanent interactions [12,27]. Thus, transient interaction partners show much less co-expression over time than permanent interaction partners [25–27].

Here, we estimated dispensable content among PPIs that are transient in time as well as PPIs that are permanent in time in human. First, we quantified gene time-course expression levels in human from 63 experiments reported in the Gene Expression Omnibus (GEO) [77], with expression levels measured over at least 5 different time points in each experiment (S6A Data). Next, we distinguished transient PPIs from permanent PPIs by measuring the temporal co-expression of interaction partners using Pearson's correlation coefficient of their gene expression profiles reported in each experiment (S7A and S7B Data). We consider a PPI to be transient in time if it is transient in the majority of experiments, where a PPI is transient in an experiment if the co-expression of its interaction partners in that experiment is less than the median co-expression of all interaction partners across all experiments (0.1 in Y2H-SI, and 0.11 in Lit-SI). According to this definition, 43% of PPIs in Y2H-SI and 43% of PPIs in Lit-SI are considered to be transient in time (S5 Data).

Given these PPI classifications, as in the previous section, we obtained from our edgotype predictions in Table 1 the probabilities for effectively neutral mutations and mildly deleterious mutations to edgetically disrupt temporally transient PPIs as well as temporally permanent PPIs in both structural interactomes, Y2H-SI and Lit-SI (Fig 3B and Table 2). Here, we also found a similarly low enrichment of PPI disruptions by neutral mutations compared to mildly deleterious mutations among both temporally transient PPIs and temporally permanent PPIs (Table 2), suggesting again that transient PPIs are as likely to be dispensable as permanent PPIs. Next, we integrated these edgotype probabilities again into our Bayesian framework described in the Methods section to calculate dispensable content among temporally transient PPIs as well as temporally permanent PPIs. As a result, we estimated that <~20% of temporally transient PPIs are completely dispensable, i.e., effectively neutral upon disruption, in both interactomes Y2H-SI and Lit-SI (Fig 3B and Table 2). We also estimated that <~20% of temporally permanent PPIs are completely dispensable in both interactomes (Fig 3B and Table 2).

## Dispensable content among spatially transient PPIs

In addition to being transient or permanent in time, PPIs can also be transient or permanent in space. A PPI is permanent in space if the two interaction partners always co-express to form stoichiometric complexes in the same tissues or cell types, and transient if otherwise [28] (Fig 3C). Here, we estimated dispensable content among PPIs that are transient in space as well as PPIs that are permanent in space in human. First, we quantified gene expression levels in 16 human body tissues using RNA-Seq gene expression data from the Illumina Body Map 2.0 project [78] (S6B Data). Next, we distinguished transient PPIs from permanent PPIs by measuring tissue co-expression of interaction partners using Pearson's correlation coefficient of their gene expression profiles (S7C and S7D Data). We consider a PPI to be transient in space if the co-expression of its interaction partners is less than the median co-expression of all interaction partners in the structural interactome (0.39 in Y2H-SI, and 0.45 in Lit-SI), otherwise we

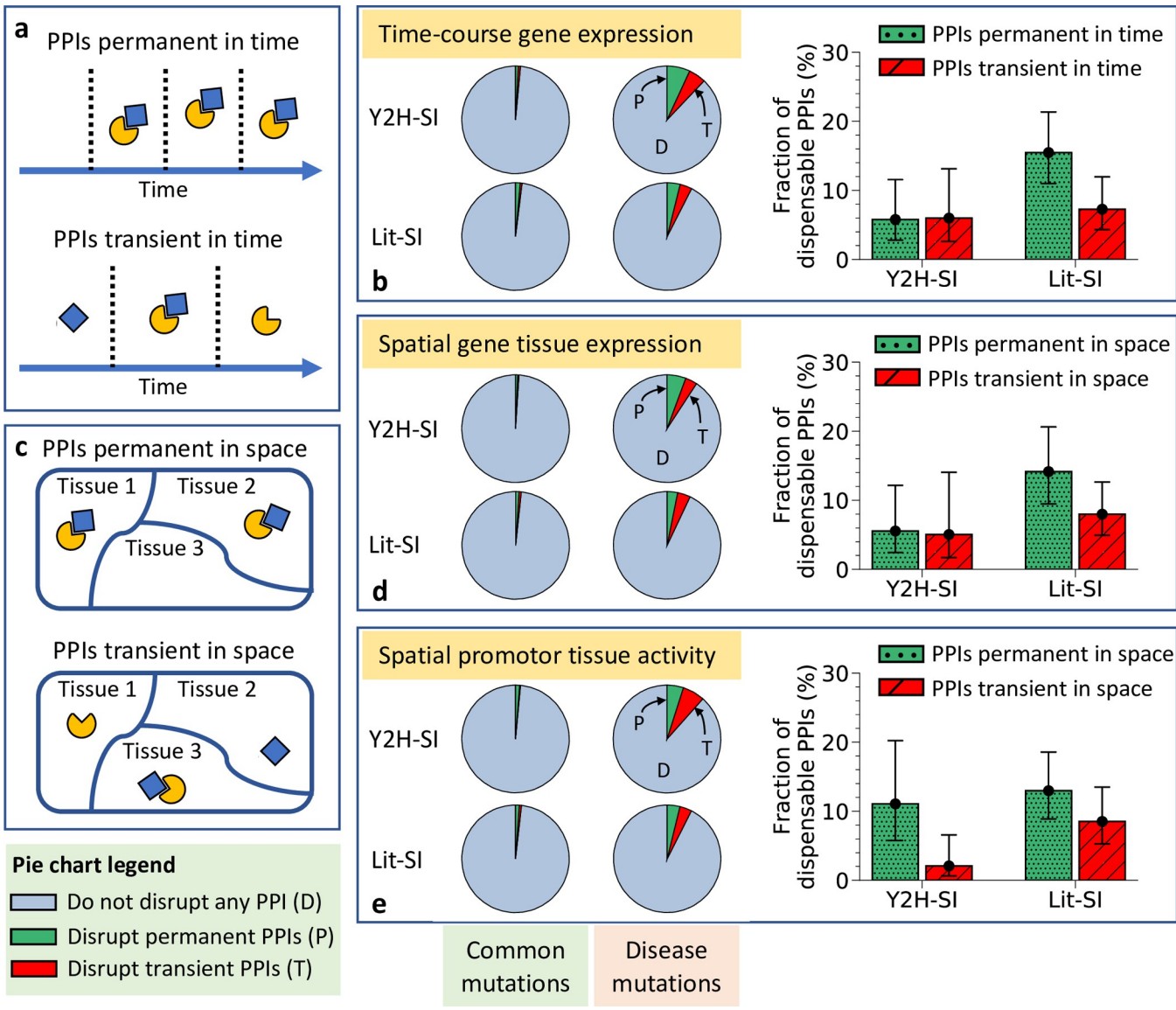

**Fig 3. Dispensable content among transient and permanent PPIs in time and space.** (a) Graphical description of temporally transient PPIs and temporally permanent PPIs. (b) Edgotype predictions (left) and dispensable content (right) among temporally transient PPIs and temporally permanent PPIs in the two human structural interactomes, Y2H-SI and Lit-SI. Transient and permanent PPIs were identified based on time-course co-expression of interaction partners derived from the Gene Expression Omnibus data. (c) Graphical description of spatially transient PPIs and spatially permanent PPIs. (d) Edgotype predictions (left) and dispensable content (right) among spatially transient PPIs and spatially permanent PPIs in the two human structural interactomes, Y2H-SI and Lit-SI. Transient and permanent PPIs were identified based on tissue co-expression of interaction partners derived from the Illumina Body Map 2.0 project data. (e) Edgotype predictions (left) and dispensable content (right) among spatially transient PPIs and spatially permanent PPIs in the two human structural interactomes, Y2H-SI and Lit-SI. Transient and permanent PPIs were identified based on correlations in gene promoter activity levels associated with interaction partners derived from the Fantom5 project data. All error bars represent 95% confidence intervals.

consider the PPI to be permanent in space. According to this definition, 49% of PPIs in Y2H-SI and 50% of PPIs in Lit-SI are considered to be transient in space (S5 Data).

Given these PPI classifications, as in previous sections, we obtained from our edgotype predictions in Table 1 the probabilities for effectively neutral mutations and mildly deleterious mutations to edgetically disrupt spatially transient PPIs as well as spatially permanent PPIs in both structural interactomes, Y2H-SI and Lit-SI (Fig 3D and Table 2). Again, we found a

similarly low enrichment of PPI disruptions by neutral mutations compared to mildly deleterious mutations among both spatially transient PPIs and spatially permanent PPIs (Table 2). Using these edgotype probabilities, we applied our Bayesian framework described in the Methods section again to estimate dispensable content among spatially transient PPIs and spatially permanent PPIs. As a result, we estimated that $<\sim 20\%$ of spatially transient PPIs are completely dispensable, i.e., effectively neutral upon disruption, in both interactomes Y2H-SI and Lit-SI (Fig 3D and Table 2). We also estimated that $<\sim 20\%$ of spatially permanent PPIs are completely dispensable in both interactomes (Fig 3D and Table 2).

In addition, we repeated our calculations again, this time distinguishing spatially transient PPIs from spatially permanent PPIs based on correlations in gene promoter activity associated with interacting proteins, as measured by the Fantom5 project in 183 human body tissue samples [79] (S6C Data). Using this promoter-level data, we consider a PPI to be transient in space if the correlation in gene promoter activity associated with its interaction partners is less than the median correlation of all interaction partners in the structural interactome (0.16 in Y2H-SI, and 0.22 in Lit-SI), otherwise we consider the PPI to be permanent in space (S7C and S7D Data). According to this definition, 50% of PPIs in Y2H-SI and 49% of PPIs in Lit-SI are considered to be transient in space (S5 Data). Given these new PPI classifications and associated edgotype predictions shown in Table 1, we also estimated that $<\sim 20\%$ of spatially transient PPIs are completely dispensable in both interactomes Y2H-SI and Lit-SI (Fig 3E and Table 2). Similarly, we estimated that $<\sim 20\%$ of spatially permanent PPIs are completely dispensable in both interactomes (Fig 3E and Table 2).

## Dispensable content among unbalanced PPIs

A fourth property that distinguishes transient PPIs from permanent PPIs is the quantitative stoichiometry of interaction [29]. While transient PPIs often involve date hubs interacting with multiple partners at different points in time [12,27], these multiple partners may often be multiple copies of the same protein [29]. Thus, unlike permanent interaction partners, transient interaction partners tend to have unbalanced ratios of abundance, with the hub protein having significantly lower expression levels than its interaction partners [29]. This unbalance in protein abundance may be observed across different points in time (Fig 4A) and/or across different tissues (Fig 4C).

Here, we estimated dispensable content among PPIs with unbalanced abundance among interaction partners as well as PPIs with balanced abundance among interaction partners, using time-course expression data as well as tissue-based expression data. First, we obtained gene time-course expression data in human from 63 experiments reported in the Gene Expression Omnibus (GEO) [77], with expression levels measured over at least 5 different time points in each experiment (S6A Data). Next, we calculated the $\log_{10}$ difference in expression levels for interaction partners at each time point in each experiment (S8 Data). We consider a PPI to be unbalanced if it is unbalanced in the majority of experiments, where a PPI is unbalanced in an experiment if the average of the absolute $\log_{10}$ difference in expression over time for its interaction partners is larger than the median value among all experiments for all interaction partners in the structural interactome (0.38 in Y2H-SI, and 0.40 in Lit-SI). According to this definition, 23% of PPIs in Y2H-SI and 26% of PPIs in Lit-SI are considered to be unbalanced over time (S5 Data).

Given these PPI classifications and our edgotype predictions in Table 1, we calculated the probabilities for effectively neutral mutations and mildly deleterious mutations to edgetically disrupt unbalanced PPIs as well as balanced PPIs in both structural interactomes, Y2H-SI and Lit-SI (Fig 4B and Table 2). As with our previous observations, we also found a similarly low

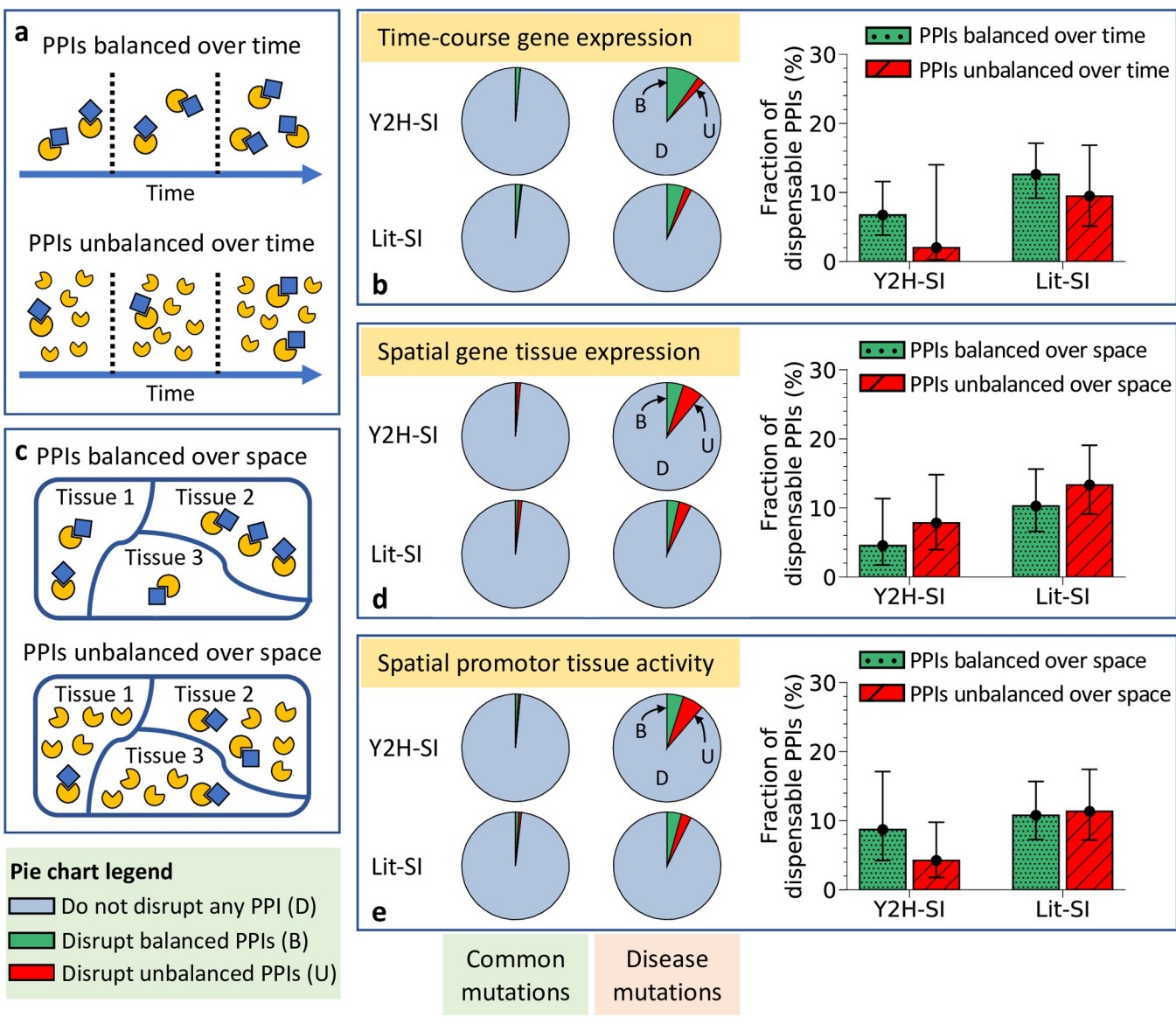

**Fig 4. Dispensable content among unbalanced and balanced PPIs over time and space.** (a) Graphical description of temporally balanced PPIs and temporally unbalanced PPIs. (b) Edgotype predictions (left) and dispensable content (right) among temporally balanced PPIs and temporally unbalanced PPIs in the two human structural interactomes, Y2H-SI and Lit-SI. Balanced and unbalanced PPIs were identified based on time-course expression levels of interaction partners derived from the Gene Expression Omnibus data. (c) Graphical illustration of spatially balanced PPIs and spatially unbalanced PPIs. (d) Edgotype predictions (left) and dispensable content (right) among spatially balanced PPIs and spatially unbalanced PPIs in the two human structural interactomes, Y2H-SI and Lit-SI. Balanced and unbalanced PPIs were identified based on tissue expression levels of interaction partners derived from the Illumina Body Map 2.0 project data. (e) Edgotype predictions (left) and dispensable content (right) among spatially balanced PPIs and spatially unbalanced PPIs in the two human structural interactomes, Y2H-SI and Lit-SI. Balanced and unbalanced PPIs were identified based on gene promoter activity levels associated with interaction partners derived from the Fantom5 project data. All error bars represent 95% confidence intervals.

enrichment of PPI disruptions by neutral mutations compared to mildly deleterious mutations among both unbalanced PPIs and balanced PPIs (Table 2). We integrated these probabilities into our Bayesian framework as in the previous sections and estimated that <~20% of unbalanced PPIs are completely dispensable, i.e., effectively neutral upon disruption, in both interactomes Y2H-SI and Lit-SI (Fig 4B and Table 2). We also estimated that <~20% of balanced PPIs are completely dispensable in both interactomes (Fig 4B and Table 2).

In addition, we also obtained gene expression levels in 16 human body tissues from the Illumina Body Map 2.0 project [78] (S6B Data), and calculated the $log_{10}$ difference in expression levels for interaction partners across all tissues (S9A and S9B Data). Using this tissue-based expression data, we consider a PPI to be unbalanced if the average of the absolute $log_{10}$ difference in expression across all tissues for its interaction partners is larger than the median value for all interaction partners in the structural interactome (0.63 in Y2H-SI, and 0.56 in Lit-SI), otherwise we consider the PPI to be balanced. According to this definition, 50% of PPIs in Y2H-SI and 50% of PPIs in Lit-SI are considered to be unbalanced over space (S5 Data). Given these PPI classifications and our edgotype predictions in Table 1, we also estimated that <~20% of both unbalanced PPIs and balanced PPIs are completely dispensable in both structural interactomes Y2H-SI and Lit-SI (Fig 4D and Table 2).

Finally, we repeated our predictions of PPI disruptions this time distinguishing balanced PPIs from unbalanced PPIs using gene promoter activity data associated with interaction partners, as measured by the Fantom5 project in 183 human body tissue samples [79] (S6C Data). Again, we consider a PPI to be unbalanced if the average of the absolute $log_{10}$ difference in expression across all tissues for its interaction partners is larger than the median value for all interaction partners in the structural interactome (0.68 in Y2H-SI, and 0.59 in Lit-SI), otherwise we consider the PPI to be balanced (S9C and S9D Data). According to this definition, also 50% of PPIs in Y2H-SI and 50% of PPIs in Lit-SI are considered to be unbalanced over space (S5 Data). Given these PPI classifications and our edgotype predictions in Table 1, we also estimated that <~20% of both unbalanced PPIs and balanced PPIs are completely dispensable in both structural interactomes (Fig 4E and Table 2).

## Dispensable content among mutually exclusive PPIs

At the structural level, one property underlying the transient or permanent nature of PPIs is the number of interaction partners targeting the same binding interface of a protein [12,27]. Date hubs use the same binding interface to carry out transient interactions with multiple partners in a mutually exclusive manner (Fig 5A). On the other hand, party hubs are able to form permanent interactions with multiple partners simultaneously using multiple interfaces (Fig 5A). Thus, we estimated dispensable content among three groups of PPIs: PPIs that do not

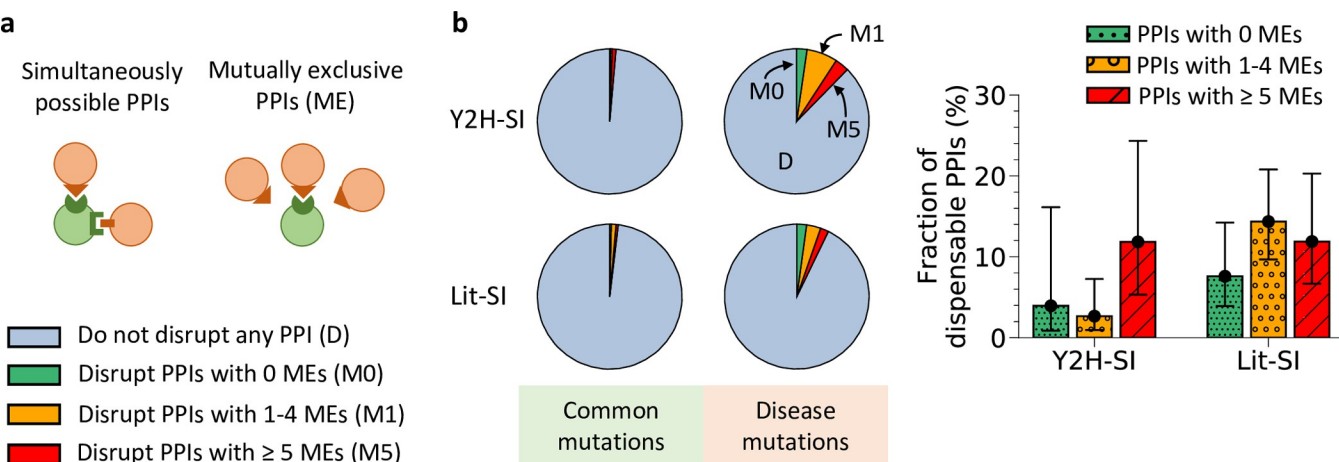

**Fig 5. Dispensable content among mutually exclusive and simultaneously possible PPIs.** (a) Graphical description of simultaneously possible PPIs mediated by a multi-interface protein and mutually exclusive PPIs mediated by a single-interface protein. (b) Edgotype predictions (left) and dispensable content (right) among PPIs that share their binding interface in a mutually exclusive manner with 0, 1 to 4, or 5 or more PPIs in the two human structural interactomes, Y2H-SI and Lit-SI. Error bars represent 95% confidence intervals.

share their binding interface with any other PPI, PPIs that share their binding interface in a mutually exclusive manner with 1 to 4 PPIs, and PPIs that share their binding interface in a mutually exclusive manner with 5 or more PPIs (S10 Data). In Y2H-SI, we found that 15% of PPIs do not share their binding interface with any other PPI, 42% share their binding interface with 1 to 4 mutually exclusive PPIs, and 43% share their binding interface with 5 or more mutually exclusive PPIs (S5A Data). In Lit-SI, we found that 16% of PPIs do not share their binding interface with any other PPI, 51% share their binding interface with 1 to 4 mutually exclusive PPIs, and 33% share their binding interface with 5 or more mutually exclusive PPIs (S5B Data).

From our edgotype predictions in Table 1, we calculated the probabilities for effectively neutral mutations and mildly deleterious mutations to disrupt PPIs belonging to either one of the three aforementioned groups in Y2H-SI and Lit-SI (Fig 5B and Table 2). Here, we also found a low enrichment of PPI disruptions by neutral mutations compared to mildly deleterious mutations among all three groups of PPIs (Table 2). Next, we integrated these probabilities into our Bayesian framework as before and calculated dispensable content among each group of PPIs. As a result, we estimated that $<\sim$20% of PPIs in each group are completely dispensable in both interactomes Y2H-SI and Lit-SI (Fig 5B and Table 2). These results reveal that dispensable content does not vary significantly with the number of mutually exclusive PPIs sharing the same binding interface.

## Discussion

In a previous study, we estimated that $<\sim$20% of PPIs in the overall human interactome are completely dispensable, that is, effectively neutral upon disruption. While this estimate represents a grand average over the entire human interactome, it remains unknown how dispensable content varies across different subsets of the interactome involved in different biological processes and pathways, and across diverse groups of PPIs that exhibit different binding patterns across time and space. In this study, we took the first step towards addressing this question, by dividing the human interactome into two major groups of PPIs, transient and permanent, and estimating dispensable content among each group using a computational approach that we had developed and validated in previous studies. Overall, we estimate that $<\sim$20% of transient PPIs in human are completely dispensable. This fraction is similar to the fraction of dispensable PPIs calculated among permanent PPIs. Our results suggest that, similar to permanent PPIs, most transient PPIs in human are important to cellular function and are subject to similarly strong selective pressures in the human interactome.

Our estimates of dispensable content among transient PPIs and permanent PPIs were derived from predicted mutation edgotypes in two human structural interactomes, Y2H-SI which was derived from the HuRI dataset consisting of PPIs recently mapped by systematic Y2H screens [68], and Lit-SI which was derived from the literature-curated dataset of PPIs reported by at least two independent experiments in the IntAct database [71]. Our estimates of dispensable content for both transient and permanent PPIs calculated in both interactomes are broadly consistent with one another ($<\sim$20%). In addition, we also estimated the dispensable content among all PPIs together in the human interactome, using predicted mutation edgotypes in Y2H-SI and Lit-SI as well as mutation edgotypes obtained from experiments of Sahni et al. 2015. These estimates are also broadly consistent with one another ($<\sim$20%). Notably, our estimates of dispensable content derived in this study from the HuRI dataset and the IntAct literature-curated dataset are consistent with our previous estimates in [20] which were derived from much smaller datasets, the HI-II-14 dataset mapped by systematic Y2H screens [72] and an earlier version of literature-curated PPIs from the IntAct database.

Our study provides a comprehensive list of transient and permanent PPIs in human characterized using diverse structural and biophysical properties. These properties include, among others, the temporal and spatial co-expression of interaction partners, a property that has not been explored enough in recent literature especially on large-scale PPI datasets such as the HuRI dataset. While discrete properties such as the number of mutually exclusive PPIs sharing the same binding interface can be strong indicators of the transient or permanent nature of interaction, other properties such as binding affinity and the spatial and temporal co-expression of interaction partners may vary for some PPIs over a continuum of values overlapping between transient and permanent PPIs. For example, while transient PPIs tend to have weak binding affinities, it may be that some of them have stronger binding affinities closer to those of permanent PPIs. At the same time, while transient PPIs tend to have low co-expression levels and unbalanced stoichiometries between interaction partners, some of them may have higher co-expression levels and more balanced stoichiometries closer to those of permanent PPIs. Hence, there is no perfect borderline that separates transient PPIs from permanent PPIs based on such properties. Our study addresses this limitation by considering different structural, biophysical and spatiotemporal properties for classifying transient and permanent PPIs. These properties produce different sets of PPI classifications that are diverse enough to capture the aforementioned variabilities in PPI properties, but also with significant overlap (S5 Data). On average, a PPI is classified as transient by ~4 out of 8 properties in both Y2H-SI and Lit-SI, with 36% of PPIs in Y2H-SI and 40% of PPIs in Lit-SI classified as transient by 5 or more properties (Fig A in S1 Text). This diversity in PPI classifications results in a noticeable change across different PPI properties in the proportions $P(T|N)$, $P(T|M)$, $P(P|N)$ and $P(P|M)$ of neutral (N) and mildly deleterious (M) mutations that edgetically disrupt transient (T) and permanent (P) PPIs (Table 2). Nonetheless, the two ratios of proportions $P(T|N)/P(T|M)$ and $P(P|N)/P(P|M)$ are less affected by such changes (Table 2), resulting in estimates of dispensable PPI content that are broadly consistent across all PPI properties (<~20%). These results highlight the importance of considering different biophysical measurements in our study for classifying transient and permanent PPIs.

The high quality of our estimates of dispensable content among both transient PPIs and permanent PPIs depends on the quality of our predictions of transient and permanent interactions. Thus, we validated the performance of our computational approach for predicting transient and permanent PPIs using multiple experimental datasets. First, we benchmarked our FoldX-based predictions of PPI binding free energy (ΔG) against experimental ΔG values obtained from the Integrated Protein-Protein Interaction Benchmarks Database [61]. We selected PPIs with experimental ΔG < -12 kcal/mol ($K_d < 10^{-9}$ M) to represent strong permanent interactions and those with experimental ΔG > -8 kcal/mol ($K_d > 10^{-6}$ M) to represent weak transient interactions [62]. We then predicted ΔG values for these two groups of PPIs using FoldX. As a result, we obtained a Pearson correlation coefficient of 0.33 between predicted and experimental ΔG (p = 3.4 x $10^{-4}$), indicating that our FoldX predictions of PPI binding free energy are high quality. Next, we predicted PPIs to be transient or permanent using ΔG values calculated by FoldX, and compared our predictions to PPI classifications based on experimental ΔG values. For predicting transient PPIs, we obtained a balanced accuracy of 68%, a precision of 90%, a true positive rate (TPR) of 0.4, and a false positive rate (FPR) of 0.03. A TPR that is equal to FPR indicates that predictions are no better than random expectations. Our TPR is 13.3 times larger than our FPR (p = 7.2 x $10^{-7}$, two-sided Fisher's exact test), further proving that our computational approach for predicting weak transient PPIs and strong permanent PPIs using FoldX-based ΔG calculations is very high-quality.

In addition to classifying PPIs based on strength of interaction, our study also classifies PPIs that are transient or permanent in time using three other properties: 1) low temporal co-

expression for transient interaction partners versus high temporal co-expression for permanent interaction partners, 2) unbalanced stoichiometry for transient interactions versus balanced stoichiometry for permanent interactions, and 3) mutually exclusive binding for transient interactions versus simultaneously possible binding for permanent interactions. Thus, we validated our predictions of transient and permanent PPIs using these three properties on a manually-curated dataset of transient and permanent complexes in human derived from two independent studies by La et al. [62] and Mintseris and Weng [63]. For predicting transient PPIs based on temporal co-expression levels, we obtained a balanced accuracy of 70%, a precision of 99%, a TPR of 0.51, and a FPR of 0.1 (Table A in S1 Text). Our TPR in this case is 5.1 times larger than our FPR (p = 0.019, two-sided Fisher's exact test), indicating that our predictions of temporally transient PPIs are very high quality and much better than random expectations. For predicting transient PPIs based on stoichiometry, we obtained a balanced accuracy of 56%, a precision of 97% and a TPR to FPR ratio of 1.4, albeit with an insignificant p-value (Table A in S1 Text). Finally, by predicting mutually exclusive PPIs to be transient and simultaneously possible PPIs to be permanent, we obtained a balanced accuracy of 78%, a precision of 98%, a TPR of 0.95, and a FPR of 0.4 (Table A in S1 Text). A TPR to FPR ratio of 2.4 (p = 1.1 x $10^{-5}$, two-sided Fisher's exact test) further proves that our computational approach for predicting temporally transient PPIs and temporally permanent PPIs is high-quality. Notably, a large part of PPI classifications in the benchmark dataset of La et al. were derived from experimental ΔG values using rigid cut-offs and may contain some errors. Therefore, the quality of our computational predictions may be even better than what is observed by validation on this dataset.

Unlike the case for PPIs that are transient or permanent in time, experimental benchmark datasets for PPIs that are transient or permanent in space are not available. This is because while transience in time can be accurately detected through protein 3D structure analysis and other single-cell data analysis, detecting transience in space across multiple body tissues and in different cell types is more challenging and time-consuming. Nonetheless, a previous computational study by Bossi and Lehner has also attempted to predict spatially transient and spatially permanent PPIs based on gene co-expression levels in 79 human body tissues [67]. Unlike our study which classifies PPIs using gene expression data from the Illumina Body Map 2.0 project and the Fantom5 project, the study of Bossi and Lehner classified PPIs using gene expression data from the GNF Atlas project [80]. While such dataset may not be accurate enough to be considered a gold standard for benchmarking our predictions of transient and permanent PPIs, it provides an independent set of PPI classifications that can be used to validate our estimates of dispensable content among spatially transient and spatially permanent PPIs. Thus, we repeated our calculations of dispensable content, this time labelling spatially transient and spatially permanent PPIs using the dataset of Bossi and Lehner. We considered a PPI to be permanent in space if it was predicted by Bossi and Lehner to exist in at least 90% of the tissues, otherwise we considered the PPI to be transient in space. In Lit-SI, we estimated that ~6% of spatially transient PPIs are completely dispensable with a 95% confidence interval of ~3–11%, and that ~8% of spatially permanent PPIs are completely dispensable with a 95% confidence interval of ~3–19%. These estimates remain below 20% in agreement with our previous estimates. A similar calculation of dispensable content in the recently mapped interactome Y2H-SI was not possible due to lack of PPI classifications in the dataset of Bossi and Lehner, which only includes PPIs that were published in the literature at the time of their study.

Our study uses gene tissue expression data from the Illumina Body Map and Fantom5 projects to classify PPIs as spatially transient or spatially permanent. This expression data was measured in normal human body tissues. Our study also uses gene time-course expression data

from the GEO database to classify PPIs as temporally transient or temporally permanent. This expression data was measured in different cell types under varying conditions. While it is very difficult to obtain detailed information about the health condition of subjects participating in these projects, it is possible that some gene expression levels may vary for individuals with certain diseases and in certain clinical settings, which may possibly introduce some errors into our classifications of transient and permanent PPIs. It is also possible that expression levels of some genes may be affected by mutations carried by the same gene. Our study assumes that gene expression samples were obtained from healthy individuals, which is true for the Illumina Body Map dataset and the Fantom5 dataset, but may not always be true for GEO data. Our study also assumes that the expression level of a gene is not affected by mutations in the gene itself. We address these limitations by considering other biophysical and structural properties of PPIs that are independent from gene expression levels, such as the strength of interaction and the number of mutually exclusive PPIs sharing the same binding interface. The consistency of our estimates of dispensable content using different PPI properties proves that our computational framework is robust to such possible sources of error. Furthermore, the high quality of our PPI classifications and estimates of dispensable PPI content as demonstrated by validations against multiple independent experimental and computational datasets of transient and permanent PPIs further proves the robustness of our study to such possible errors in PPI classifications. While mutation profiles associated with the gene expression samples that were used in this study are not available, sample information provided in our supplementary files may be used in the future for further investigation of mutation impact on gene expression.

Our structure-based edgotype prediction approach predicts that a mutation at the PPI binding interface edgetically disrupts the PPI if and only if it causes a change in binding free energy $\Delta\Delta G > 0.5$ kcal/mol. This $\Delta\Delta G$ cut-off has been used and proven to provide high-quality predictions of PPI disruption in previous structural biology studies [20,21,81,82]. In addition, our Bayesian framework for estimating dispensable content has been shown to be robust to different choices of $\Delta\Delta G$ cut-off close to 0.5 kcal/mol, particularly 0.3 and 0.7 kcal/mol [21]. Nonetheless, it remains possible that some strong interactions require a change in binding energy significantly larger than our cut-off of 0.5 kcal/mol to be disrupted. In such cases, a mutation that causes a change in binding energy that is only slightly larger than 0.5 kcal/mol may be falsely predicted to disrupt the PPI by our computational approach. It is worth noting here that common mutations tend to have smaller $\Delta\Delta G$ values compared to disease mutations, with an average of 0.33 and 0.28 kcal/mol for common mutations in Y2H-SI and Lit-SI, respectively, compared to 1.1 and 0.9 kcal/mol for disease mutations, respectively (Fig B in S1 Text). Therefore, using a higher $\Delta\Delta G$ cut-off for predicting edgetic mutations is more likely to reduce the proportion of neutral mutations that are edgetic by a larger fraction compared to deleterious mutations, resulting in estimates of dispensable PPI content that are even lower than our current estimates. Indeed, we repeated our edgotype predictions and re-calculated dispensable content among both weak and strong PPIs using significantly higher $\Delta\Delta G$ cut-offs for predicting edgetic mutations: 1, 2 and 3 kcal/mol, and our estimates of dispensable content among both weak PPIs and strong PPIs remain below ~20% (Table B in S1 Text).

In our study, we used the FoldX method to calculate the change in PPI binding free energy upon mutation. Other computational methods are also available [83]. While FoldX uses only physics-based calculations to predict $\Delta\Delta G$, other methods often make use of protein sequence and evolutionary information [83], which may introduce biases into our edgotype predictions for deleterious and neutral mutations. Furthermore, very few methods offer the option of predicting $\Delta\Delta G$ for thousands of mutations simultaneously in reasonable time like FoldX. Nonetheless, our previous studies have shown that our Bayesian framework for estimating dispensable PPI content and for estimating the overall fitness effect for different mutation

edgetypes is robust to different choices of methods for predicting ΔΔG upon mutation [20,21], including the FoldX method [75], BindProfX [84], mCSM-PPI2 [85] and DynaMut2 [86]. At the same time, FoldX has been shown to provide high-quality predictions of binding ΔΔG in previous studies, with Pearson correlation coefficients ranging from 0.4 to 0.5 when bench-marked against comprehensive datasets of experimentally-determined ΔΔG values [83]. FoldX particularly outperforms other methods in identifying disease mutations [87]. An independent experiment performed by our recent study [20] reported Pearson correlation coefficients of 0.5 and 0.42 for ΔΔG values predicted by FoldX on co-crystal structures and homology models, respectively, when benchmarked against experimental data in the SKEMPI database [88]. In another study [21], we validated our FoldX-based edgetype prediction method on the experimental data of Sahni et al. [4] using a ΔΔG cut-off of 0.5 kcal/mol for predicting edgetic PPI disruption by mutation at the binding interface. Out of 23 mutations that were found to be edgetic in experiments, 7 were correctly predicted by our method to be edgetic, giving a true positive rate (TPR) of 0.3. On the other hand, out of 57 mutations that were found to be non-edgetic by experiments, only 2 were incorrectly predicted by our method to be edgetic, giving a false positive rate (FPR) of 0.04. A TPR that is equal to FPR indicates that predictions are no better than random expectations. Our TPR is 7.5 times larger than our FPR (p = 0.002, two-sided Fisher's exact test), confirming again that our FoldX-based method for predicting edgetic PPI disruptions is very high-quality.

To further validate the high quality of our predicted mutation edgetypes and the resulting estimates of dispensable content, we estimated dispensable content for both transient and permanent PPIs this time using mutation edgetypes obtained from experiments of Sahni et al.. We first classified PPIs in the experimental dataset as either transient or permanent, in both time and space, as well as balanced or unbalanced in stoichiometry over both time and space, following the same procedures we used to classify PPIs in Y2H-SI and Lit-SI. It is worth noting here the very small size of the experimental dataset which consists of only 47 common mutations and 197 disease mutations, with only 2 common mutations and 62 disease mutations that are edgetic. Next, we calculated the fractions of mutations that are edgetic among transient PPIs (transient in time, transient in space, and unbalanced in stoichiometry) and the fraction of mutations that are edgetic among permanent PPIs (permanent in time, permanent in space, and balanced in stoichiometry), using edgetype data from experiments. Similar to our calculations in Y2H-SI and Lit-SI, we found that transient and permanent PPIs are both less likely to be disrupted by neutral mutations than by deleterious mutations (Tables C and D in S1 Text). We then estimated dispensable content among both transient PPIs and permanent PPIs using these edgetype results derived from experiments. Overall, our estimates are all below ~20%, consistent with our previous estimates obtained from predictions in Y2H-SI and Lit-SI, albeit with larger confidence intervals due to small sample size. These results derived from experimental mutation edgetypes further prove the high quality of our predicted mutation edgetypes and dispensable content estimates obtained from these predictions. A similar calculation of dispensable content among weak PPIs and strong PPIs and among mutually exclusive PPIs and simultaneously possible PPIs was not possible due to lack of structural data for the vast majority of PPIs that are disrupted by mutations in experiments.

Our calculations make a clear distinction between edgetic mutations and quasi-null mutations. While edgetic mutations disrupt specific PPIs by disrupting only the binding interface, quasi-null mutations disrupt all PPIs by disrupting overall protein stability thus creating other complex cellular and phenotypic changes that cannot be explained by simple PPI disruption. This distinction is not as simple in the experimental study of Sahni et al. due to lack of structural information. There, a mutation is considered to be edgetic if it disrupts at least one PPI but not all PPIs associated with a protein, and considered to be quasi-null if it disrupts all PPIs.

This definition is not completely accurate since some edgetic mutations may disrupt all PPIs by disrupting the binding interface while maintaining overall protein stability, and will be misclassified as quasi-null. However, calculations in our previous study show that treating quasi-null mutations from experiments as if they were edgetic has a negligible impact on our estimate of dispensable content in the overall human interactome [20], indicating that our estimate of dispensable content derived from experiments is robust to such possible errors. At the same time, our calculations assume that each edgetic mutation disrupts one PPI, which is true for most edgetic mutations in Y2H-SI (50% of common mutations, and 58% of disease mutations) and in Lit-SI (82% of common mutations, and 66% of disease mutations). Nonetheless, to check whether our estimates of dispensable content are robust to the existence of edgetic mutations that disrupt more than one PPI, we repeated our calculations of dispensable content among both transient PPIs and permanent PPIs, this time replacing the fraction of mutations that are edgetic among common mutations and among disease mutations with the fraction of mutations that are mono-edgetic, i.e., those that disrupt only one PPI. Our results show that dispensable content among both transient and permanent PPIs remains below ~20% (Tables E and F in S1 Text), albeit with larger confidence intervals in some cases due to smaller sample size for mono-edgetic mutations (Table G in S1 Text).

Our estimates of dispensable PPI content are also robust to the presence of experimental false positives ("erroneous PPIs") in PPI datasets [55,89]. These false positives mostly include physical interactions that are detected *in vitro* but do not occur *in vivo*, indirect interactions between proteins within the same complex that do not interact directly, as well as other stochastic artifacts that cannot be reproduced by independent experiments. Our structure-based approach includes several measures to minimize such false positive errors. First, we started with PPIs obtained from experiments rather than predictions. Second, the HuRI dataset was subjected to multiple Y2H screens and other quality control measures, and is similar in quality to a gold-standard dataset of literature-derived PPIs [68]. In addition, our IntAct-derived dataset includes only high-quality PPIs reported by at least two independent experiments in the literature. Furthermore, our structural interactomes include only PPIs for which we were able to construct homology models using experimentally determined 3D structural templates in PDB. Thus, our homology modelling approach enriches for true physical interactions and minimizes the occurrence of false positives.

Despite these quality control measures, it remains a possibility that some false positives may exist in our structural interactomes. In the presence of such errors, our estimates of dispensable content among both transient PPIs and permanent PPIs represent upper bounds. This is because erroneous PPIs have no biological power to discriminate between neutral and deleterious mutations. Thus, in the false positive portion of the PPI dataset, the probability for an edgetic mutation that disrupts a PPI to be effectively neutral is independent of the mutation edgotype and is similar to the prior probability for any missense mutation to be effectively neutral, which is ~27%. Our estimates of dispensable content among both transient PPIs and permanent PPIs are defined as the probability for an edgetic mutation that disrupts such PPIs to be effectively neutral. These estimates represent the average calculated over the mixed dataset of true PPIs and erroneous PPIs, and are both lower than ~20%. Therefore, the fraction of dispensable PPIs calculated in the error-free portion of the dataset will be even lower than our average estimates.

The prior probabilities P(N), P(M) and P(S) were obtained from a genome-wide population genetics study that is completely independent from our edgotype predictions [76]. These prior probabilities are high-quality, as they were subjected to multiple quality-control measures. Nonetheless, it remains a possibility that the error margins associated with these probabilities are not negligible. Our conclusions are also robust to such possible errors. From Eq 1 in the

Methods section, it is clear that the fraction of type-T PPIs that are effectively neutral (N) upon edgetic disruption P(N|T) depends only on the two ratios P(M)/P(N) and P(T|M)/P(T|N) as follows:

$$\frac{1}{P(N|T)} = 1 + \frac{P(M)}{P(N)} \cdot \frac{P(T|M)}{P(T|N)}$$

Our upper limit of ~20% based on the 95% confidence intervals for dispensable content among both transient PPIs and permanent PPIs corresponds to the ratios P(M)/P(N) = ~2 and P(T|M)/P(T|N) = ~2. If instead of the literature-derived priors, we assume uninformative priors where P(M)/P(N) = 1, our upper limit on dispensable content will only increase from ~20% to ~33%, and we still conclude that most transient PPIs in human are indispensable. Finally, since we used the same prior probabilities to estimate the dispensable content for both transient and permanent PPIs, different choices of prior probabilities will not change our conclusion that the dispensable content for transient interactions is similar to that for permanent interactions. Our conclusion is driven primarily by the observation that while disease mutations are significantly more likely to edgetically disrupt PPIs than common mutations, the propensity for common mutations to disrupt PPIs relative to disease mutations is roughly the same among both transient PPIs and permanent PPIs. These propensities are independent of the prior probabilities used in our Bayesian framework for calculating dispensable PPI content.

The ideal way of calculating dispensable content among transient or permanent PPIs is to first perform large-scale experiments to determine whether each PPI in the human interactome is transient or permanent. Such experiments involve many challenges such as measuring the binding affinity and/or the duration of each interaction, determining whether each interaction exists or does not exist in each body tissue, and also monitoring the stoichiometry for each pair of interacting proteins over the course of interaction. The second step would be to systematically disrupt PPIs one at a time and measure the fitness change of the cell in response to each disruption. In the absence of such challenging experiments, our computational approach offers the next best solution by first classifying transient and permanent interactions using structure-based calculations as well as protein abundance measurements, predicting PPI disruptions by mutations using structure-based calculations, and finally examining the phenotypic consequences of mutations disrupting as few as one PPI at a time while maintaining all other aspects of cell biology such as protein stability, protein expression, and other molecular interactions. All these steps in our computational framework have proven to be very high-quality when benchmarked against multiple experimental datasets and therefore greatly complement experimental efforts.

In summary, we estimate that $<$~20% of transient PPIs in the human structural interactome are completely dispensable, similar to permanent PPIs, suggesting that most transient PPIs in the human structural interactome carry out important cellular functions and are at least mildly deleterious upon disruption. This estimate represents an average over all transient PPIs in the structural interactome and is likely to vary significantly across the entire human interactome. For example, dispensable content may be higher among transient PPIs mediated specifically by motif-domain interactions in intrinsically disordered regions [33,34]. Selective pressures may also be lower among transient PPIs mediated by protein domains from recently much expanded families compared to PPIs mediated by more conserved domains [90]. While our study is only concerned about estimating the average fraction of dispensable PPIs among all transient PPIs in the human structural interactome, it remains to be seen whether the dispensable content varies significantly for these different groups of transient PPIs in the entire

human interactome. At the same time, PPIs that are completely dispensable across closely related species are expected to be less likely to be conserved across these species. That being said, PPIs that are completely dispensable in human may not be dispensable in other closely related species such as chimp and mouse, and as such they may or may not be conserved among closely related species. With currently incomplete PPI experimental datasets (with high false negative rates), it is hard to determine whether a specific PPI that is dispensable in human is conserved or not in another species. Nonetheless, recent genome-wide screens suggest that ~50% of PPIs in human are lost in budding yeast [9]. Our current study suggests that at most ~20% of PPIs are completely dispensable in human (for both transient and permanent PPIs). Taken together, these results suggest that at least ~30% of PPIs are rewired between human and yeast not because they are unimportant in both species, but rather due to other factors such as lineage-specific adaption. While it has been shown that transient PPIs are more likely to rewire during evolution than permanent PPIs [12,35,50–52], these rewiring events may be driven by many complex evolutionary forces, including both non-adaptive (e.g., genetic drift on dispensable PPIs) and adaptive (e.g., lineage-specific adaptation) processes. In addition, while the removal of a single dispensable PPI has no impact on organismal fitness, in some cases it may decrease the organism's robustness against further disruptions of the interactome, for example when an important biological function is carried out by multiple redundant PPIs. Thus, the relationship between dispensability and evolutionary rate is a complex and significant research topic in and of itself, previously only investigated for proteins [91–93], but never for PPIs. While our current study is focused on PPI dispensability in human, the relationship between dispensability, its underlying molecular mechanisms and evolutionary rate for PPIs should be further investigated in future work.

## Methods

### Building the human structural interactome

Three-dimensional (3D) protein complex structures at atomic resolution were obtained from PDB [60]. For structures containing more than one model, the first model was selected. Gene Ensembl IDs in the HuRI reference interactome were mapped to protein UniProt IDs and corresponding amino acid sequences using the ID mapping table provided by UniProt [94]. For proteins in the IntAct reference interactome, UniProt IDs provided by the IntAct database were used to obtain corresponding amino acid sequences. Next, we used BLAST [95] to perform sequence alignment of all protein sequences against all PDB chain sequences found in PDB's SEQRES records, with an E-value cut-off of $10^{-5}$. For each pair of protein sequence and PDB chain, the alignment with the smallest E-value was retained, and the remaining alignments were discarded. A PPI was annotated with a pair of chains found in the same PDB structure if: (i) the two chains had a binding interface, (ii) one of the proteins in the PPI has a sequence alignment with one of the chains in the chain pair with ≥50% of interface residues mapped onto the protein; and (iii) the other protein in the PPI has a sequence alignment with the other chain in the chain pair with ≥50% of interface residues mapped onto the protein. PPIs having no PDB chain-pair annotations were discarded. The 3D structure corresponding to the annotated chain-pair of each PPI was selected as a template for generating the PPI structural model. We then used BLAST again to generate the sequence alignment for each PPI against residues with 3D coordinates available in the template file. These alignments were then used to construct PPI structural models with the MODELLER library (version 9.23) [96]. Interface residues for each PPI were identified by calculating the pair-wise Euclidean distance between residues across the two chains in the structural model. The distance between two residues was calculated as the minimum distance between all atoms of the first residue and all

atoms of the second residue. Residues in each chain that are within a distance of 5Å from any residue in the other chain were labelled as interface residues.

### Processing disease mutations

Germline mutations in human with associated phenotypic consequences were retrieved in February 2020 from the ClinVar database (genome assembly GRCh38) [73]. We selected missense mutations that are strictly labelled as pathogenic only, with supporting evidence (i.e., with at least one star), and with no conflicting phenotypic interpretations.

### Processing common mutations

Single Nucleotide Polymorphisms (SNPs) in human were retrieved in February 2020 from the Single Nucleotide Polymorphism Database (dbSNP) (build 150 GRCh38p7) [74]. First, we selected only missense mutations that are labelled as validated and not withdrawn, and are assigned a location on the RefSeq transcript of a protein. Next, we discarded all mutations labelled with disease assertions (e.g., pathogenic, likely pathogenic, drug-response, uncertain significance or other). Finally, we selected mutations that have minor allele frequencies $\geq 1\%$, as common mutations with high frequencies are unlikely to be associated with any disease.

### Mapping mutations onto the human structural interactome

We searched the protein RefSeq transcript associated with each mutation for the mutation flanking sequence, defined as either the first 10 amino acid residues or all amino acid residues, whichever one is shorter, on both sides of the mutation. Then we searched the protein's sequence designated by UniProt for the mutation flanking sequence obtained from the RefSeq transcript. If the flanking sequence was found on the protein UniProt sequence at the same position reported on the RefSeq transcript, the mutation was retained for further analysis, otherwise the mutation was discarded. For multiple mutations mapping onto the same position, only one mutation was retained for further analysis. Common mutations overlapping in position with disease mutations were also discarded. Finally, mutations located at PPI binding interfaces were identified using residue position mappings between protein UniProt sequences and PPI structural models.

### Calculating change in PPI binding free energy upon mutation

PPI structural models were first repaired using the RepairPDB command in FoldX. Change in PPI binding free energy ($\Delta\Delta G$) was then calculated for each interfacial mutation on the repaired structural model using the BuildModel command in FoldX with default parameters (temperature = 298, pH = 7.0, ionStrength = 0.05, water = -IGNORE, vdwDesign = 2).

### Calculating PPI binding free energy

PPI structural templates were first repaired using the RepairPDB command in FoldX. Binding free energy ($\Delta G$) was then calculated on repaired PPI structural templates using the AnalyseComplex command in FoldX with default parameters (temperature = 298, pH = 7.0, ionStrength = 0.05, water = -IGNORE, vdwDesign = 2).

### Processing gene tissue expression profiles

Gene tissue expression data in human was retrieved from two databases: the Illumina Body Map 2.0 project with RNA-Seq data quantified in 16 normal human body tissues [78], and the Fantom5 project with CAGE (Cap Analysis of Gene Expression) peaks (tags per million) for

gene promoters in 183 normal human body tissue samples [79]. For Illumina Body Map data, gene expression profiles were paired with proteins in the structural interactome by mapping gene names to protein UniProt IDs using UniProt's ID mapping table. For Fantom5 data, promoter CAGE peaks were paired with proteins in the structural interactome by mapping gene HGNC IDs to protein UniProt IDs using UniProt's ID mapping table. For genes with multiple CAGE peaks, the average over all peaks was considered. Tissue co-expression levels for pairs of proteins were then calculated using Pearson's correlation coefficient of their gene tissue expression profiles. Only protein pairs whose expression levels are defined together in at least 5 tissues were considered.

## Processing gene time-course expression profiles

Gene time-course expression data in human was retrieved from the Gene Expression Omnibus [77] by searching the database for curated datasets having the term 'time course' in their title or description. In total, we obtained 223 datasets for human. We then selected 63 datasets that have gene expression levels measured over at least 5 time points, with multiple samples averaged for each time point. Gene expression profiles in each experiment were paired with proteins in the structural interactome by mapping gene names to protein UniProt IDs using UniProt's ID mapping table. Time-course co-expression for pairs of proteins in each experiment was then calculated using Pearson's correlation coefficient of their gene expression profiles.

## Calculating the fraction of completely dispensable PPIs

Each mutation either edgetically disrupts a PPI of type T or does not edgetically disrupt a PPI of type T. In addition, the fitness effect of a mutation can be either neutral, mildly deleterious, or strongly detrimental. From mutation edgotype data, we obtain the probability P(T|N) for neutral (N) mutations to edgetically disrupt a type-T PPI and the probability P(T|M) for mildly deleterious (M) mutations to edgetically disrupt a type-T PPI. Furthermore, we obtain from Kryukov et al. [76] the probabilities for new missense mutations to be effectively neutral (N), mildly deleterious (M), or strongly detrimental (S): P(N) = 27%, P(M) = 53%, P(S) = 20%. We then integrate these probabilities together to calculate the probability for a new missense mutation to edgetically disrupt a type-T PPI:

$$P(T) = P(T|N)P(N) + P(T|M)P(M) + P(T|S)P(S) \qquad \text{(Eq 1)}$$

where P(T|S) ≈ 0 assuming that strongly detrimental mutations are quasi-null rather than edgetic. Finally, we apply Bayes' theorem P(A|B) = P(B|A)P(A)/P(B) to calculate the fraction of type-T PPIs that are completely dispensable, defined as the probability for a mutation that edgetically disrupts a type-T PPI to be effectively neutral (N):

$$P(N|T) = \frac{P(T|N)P(N)}{P(T)} \qquad \text{(Eq 2)}$$

Below, we describe the procedure for calculating the 95% confidence interval for P(N|T).

By substituting P(T) = P(T|N)P(N) + P(T|M)P(M) from Eq 1 into Eq 2, it is easy to see that P(N|T) only depends on the ratio P(T|M)/P(T|N) in the following way:

$$\frac{1}{P(N|T)} = 1 + \frac{P(T|M)}{P(T|N)} \cdot \frac{P(M)}{P(N)} \qquad \text{(Eq 3)}$$

where P(M)/P(N) is a constant. The 95% confidence interval for the ratio of two proportions P

(T|M)/P(T|N) was calculated according to Bland [97], which was then used to calculate the 95% confidence interval for P(N|T) using Eq 3.

## Supporting information

**S1 Data. The human structural interactome.** PPIs in the human structural interactome annotated with interface residue positions on canonical protein sequences from UniProt. (A) Y2H-SI. (B) Lit-SI.
(XLSX)

**S2 Data. Edgetic PPI disruptions and mutation edgotypes.** (A) Common mutations in Y2H-SI. (B) Disease mutations in Y2H-SI. (C) Common mutations in Lit-SI. (D) Disease mutations in Lit-SI.
(XLSX)

**S3 Data. PPI change in binding free energy.** PPI change in binding free energy ($\Delta\Delta G$) upon mutation calculated on PPI structural models. (A) Common mutations in Y2H-SI. (B) Disease mutations in Y2H-SI. (C) Common mutations in Lit-SI. (D) Disease mutations in Lit-SI.
(XLSX)

**S4 Data. PPI binding free energy.** PPI binding free energy ($\Delta G$) calculated on PPI structural templates from PDB.. (A) Y2H-SI. (B) Lit-SI.
(XLSX)

**S5 Data. PPI classifications.** (A) Y2H-SI. (B) Lit-SI.
(XLSX)

**S6 Data. Gene expression sample information.** (A) Sample counts and time points for all 63 datasets from GEO. (B) Names of the 16 tissues in the Illumina Body Map 2.0 dataset. (C) Names of the 183 tissue samples in the Fantom5 dataset.
(XLSX)

**S7 Data. Co-expression of interaction partners.** (A) Time-course co-expression in 63 GEO datasets for PPIs in Y2H-SI. (B) Time-course co-expression in 63 GEO datasets for PPIs in Lit-SI. (C) Tissue co-expression based on Illumina Body Map and Fantom5 data for PPIs in Y2H-SI. (D) Tissue co-expression based on Illumina Body Map and Fantom5 data for PPIs in Lit-SI.
(XLSX)

**S8 Data. Time-course abundance of interaction partners.** Difference in the $\log_{10}$ of time-course expression levels for interaction partners in the human structural interactome. (A) Expression difference at each time point in 63 GEO datasets for PPIs in Y2H-SI. (B) Expression difference at each time point in 63 GEO datasets for PPIs in Lit-SI.
(XLSX)

**S9 Data. Tissue-based abundance of interaction partners.** Difference in the $\log_{10}$ of tissue-based expression levels for interaction partners in the human structural interactome. (A) Expression difference among 16 tissues from Illumina Body Map for PPIs in Y2H-SI. (B) Expression difference among 16 tissues from Illumina Body Map for PPIs in Lit-SI. (C) Expression difference among 183 tissue samples from Fantom5 for PPIs in Y2H-SI. (D) Expression difference among 183 tissues samples from Fantom5 for PPIs in Lit-SI.
(XLSX)

**S10 Data. Mutually exclusive and simultaneously possible PPIs.** (A) Y2H-SI. (B) Lit-SI. (XLSX)

**S1 Text. Fig A in S1 Text. Distribution of PPI transient classifications across multiple properties**. Distribution of the number of times a PPI was classified as transient based on 8 different structural and biophysical measurements: PPI strength, transience in time (based on gene expression levels), transience in space (based on gene expression and promotor activity levels), stoichiometry in time (based on gene expression levels), stoichiometry in space (based on gene expression and promotor activity levels), and number of mutually exclusive PPIs.

**Fig B in S1 Text. Change in PPI binding free energy upon mutation**. Change in PPI binding free energy ($\Delta\Delta G$) distribution for all disease and common non-disease interfacial mutations in both structural interactomes Y2H-SI and Lit-SI. $\Delta\Delta G$ values were calculated using FoldX.

**Table A in S1 Text. Performance of predictions of temporally transient and permanent PPIs**. Performance measures for predicting temporally transient PPIs and temporally permanent PPIs in human when benchmarked against experimental data combined from La et al. (2013) and Mintseris and Weng (2003).

**Table B in S1 Text. Dispensable content derived from different binding $\Delta\Delta G$ cut-offs**. Dispensable content among weak and strong PPIs calculated using different binding $\Delta\Delta G$ cut-offs for predicting edgetic disruptions of PPIs in the two human structural interactomes Y2H-SI and Lit-SI.

**Table C in S1 Text. Edgetic mutation data derived from experiments**. Number of common mutations and disease mutations in the experimental data of Sahni et al. 2015 that edgetically disrupt transient and permanent PPIs as defined by different biophysical and spatiotemporal properties. PPI disruptions by mutations were obtained from experiments of Sahni et al. 2015. PPI classifications were determined computationally.

**Table D in S1 Text. Dispensable content among transient and permanent PPIs in experiments**. Edgotype probabilities for neutral and mildly deleterious mutations in the experimental data of Sahni et al. 2015 calculated among transient and permanent PPIs directly from Table C, assuming that common mutations are effectively neutral (N) and that disease mutations are mildly deleterious (M) on average. The resulting dispensable contents P(N|T) and P(N|P) among both transient (T) and permanent (P) PPIs were calculated using Bayes' theorem. Columns represent the following, P(T|N): probability (%) for neutral mutations (N) to edgetically disrupt transient PPIs (T), P(T|M): probability (%) for mildly deleterious mutations (M) to edgetically disrupt transient PPIs (T), P(N|T): dispensable content among transient PPIs defined as the probability (%) for transient PPIs to be effectively neutral upon disruption, P(P|N): probability (%) for neutral mutations (N) to edgetically disrupt permanent PPIs (P), P(P|M): probability (%) for mildly deleterious mutations (M) to edgetically disrupt permanent PPIs (P), P(N|P): dispensable content among permanent PPIs defined as the probability (%) for permanent PPIs to be effectively neutral upon disruption. CI: 95% confidence interval (%) for dispensable contents P(N|T) and P(N|P). PPI disruptions by mutations were obtained from experiments of Sahni et al. 2015. PPI classifications were determined computationally.

**Table E in S1 Text. Mono-edgetic mutation data obtained from predictions**. Number of common mutations and disease mutations that edgetically disrupt a single transient PPI or

permanent PPI defined by different structural, biophysical and spatiotemporal properties in the two human structural interactomes Y2H-SI and Lit-SI.

**Table F in S1 Text. Dispensable content among transient and permanent PPIs based on mono-edgetic mutations**. Edgotype probabilities for neutral and mildly deleterious mutations calculated directly from edgotype numbers in Table E, assuming that common mutations are effectively neutral (N) and that disease mutations are mildly deleterious (M) on average. The resulting dispensable contents P(N|T) and P(N|P) among both transient (T) and permanent (P) PPIs were calculated using Bayes' theorem. Columns represent the following, SI: structural interactome, P(T|N): probability (%) for neutral mutations (N) to edgetically disrupt a single transient PPI (T), P(T|M): probability (%) for mildly deleterious mutations (M) to edgetically disrupt a single transient PPI (T), P(N|T): dispensable content among transient PPIs defined as the probability (%) for transient PPIs to be effectively neutral upon disruption, P(P|N): probability (%) for neutral mutations (N) to edgetically disrupt a single permanent PPI (P), P(P|M): probability (%) for mildly deleterious mutations (M) to edgetically disrupt a single permanent PPI (P), P(N|P): dispensable content among permanent PPIs defined as the probability (%) for permanent PPIs to be effectively neutral upon disruption.

**Table G in S1 Text. Confidence intervals for dispensable content based on mono-edgetic mutations**. 95% confidence intervals for estimates of dispensable content P(N|T) and P(N|P) in Table F based on mono-edgetic mutations.
(PDF)

## Author Contributions

**Conceptualization:** Mohamed Ali Ghadie, Yu Xia.

**Data curation:** Mohamed Ali Ghadie.

**Formal analysis:** Mohamed Ali Ghadie.

**Funding acquisition:** Yu Xia.

**Investigation:** Mohamed Ali Ghadie.

**Methodology:** Mohamed Ali Ghadie, Yu Xia.

**Project administration:** Yu Xia.

**Resources:** Yu Xia.

**Software:** Mohamed Ali Ghadie.

**Supervision:** Yu Xia.

**Validation:** Mohamed Ali Ghadie.

**Visualization:** Mohamed Ali Ghadie.

**Writing – original draft:** Mohamed Ali Ghadie.

**Writing – review & editing:** Mohamed Ali Ghadie, Yu Xia.

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
