## [Decision Letter · Decision Letter 0]

10 Sep 2021

Dear Dr. Xia,

Thank you very much for submitting your manuscript "Are transient protein-protein interactions more dispensable?" for consideration at PLOS Computational Biology.

As with all papers reviewed by the journal, your manuscript was reviewed by members of the editorial board and by several independent reviewers. In light of the reviews (below this email), we would like to invite the resubmission of a significantly-revised version that takes into account the reviewers' comments. 

We cannot make any decision about publication until we have seen the revised manuscript and your response to the reviewers' comments. Your revised manuscript is also likely to be sent to reviewers for further evaluation.

Sincerely,

Ozlem Keskin

Associate Editor

PLOS Computational Biology

Arne Elofsson

Deputy Editor

PLOS Computational Biology

Reviewer's Responses to Questions

**Comments to the Authors:**

Reviewer #1: The review is uploaded as an attachment.

Reviewer #2: The authors present an analysis of the human interactome according to their dispensability and whether they’re transient or permanent. The dispensability is estimated using a Bayesian framework the authors introduced in previous work and based on mapping different types of SNPs onto the transcriptome. Transient vs. permanent is investigated in a number of different ways, including interaction strength, transience in time and space, etc.. Their main finding is that dispensability is at <20% and broadly similar for transient/permanent interactions. It is an interesting and counterintuitive finding (see below).

Overall, I think this is an interesting study, making use of fairly recently published large-scale data.

A number of points to be addressed:

1) The result that neither interaction strength nor permanence in time/space (or mutual exclusivity) is a factor for dispensability is quite counterintuitive and perhaps should be discussed more, in particular from an evolutionary standpoint. One would think that selection pressure would be lower on several kinds of transient interactions, e.g., those between protein domains from recently much expanded families and its partner, than on very old and tight protein complexes.

2) Also, if they were fully dispensable, there should also be fixed mutations (in chimp/mouse/etc.) that would get interfere with these interactions?

3) I can see that there is relatively little data on Kds for PPIs, but I’m not sure how reliable FoldX is for absolute Kds. Perhaps at least a bit of benchmarking would be in order?

4) Along similar lines, a ddG of 0.5 seems to be a fairly low cutoff. In particular if a dG of -25 is used as the cutoff for “strong”. Would it not mean that most “strong” interactions are hardly affected by some of the mutations? (going from, say, -30 kcal/mol to -29.5 kcal/mol)

Reviewer #3: Ghadie and Xua analyze the dispensable content among the transient protein-proteins interactions (PPIs) by comparing them to permanent PPIs. They map PPIs to the structural protein interactome and consider them together with the neutral and disease-associated mutations. The overarching aim of the study is stated as assessment of the importance of transient PPIs quantitatively. For this aim, authors collect data from several resources, including high-throughput PPI data, tissue specific expression data, time series data and integrates them to label interactions as transient and permanent. Additionally, they compare these two interaction types based on several characteristics. Although the study has important analysis addition, I have major concerns on the novelty that should be addressed in the manuscript. I have divided my comments into the sections to easily follow:

Introduction:

1. Introduction section has a very limited number of references despite there are many works studying transient interactions both computationally and experimentally. For example, between line numbers 53-65 authors have many strong statements, but none of them are supported with any citation. These statements need to be justified and properly supported with references.

2. “The question of how important transient PPIs…… ” sentence in line 62 sounds like still the importance of the transient PPIs not discovered. However, many studies either experimentally or computationally demonstrated that transient PPIs are an important component of signaling pathways, membrane interactions. Additionally, many experimental works are present and going on to quantitively measure the transient PPIs. I would suggest authors to include a review of experimental efforts and why computational approaches are important in this sense.

3. In the last paragraph of the Introduction, authors summarize their study. However, it is very long, and many technical details are included which are already there in the Results. This makes the paragraph very diffuse. Additionally, authors describe how they constructed the structural interactome between lines 85-90 which is the same procedure of a work by the same authors (Ghadie and Xua 2019, Nat Comm.). Repeating this part reduces the novelty aspect of the work. Is there any significant difference, update or improvement in the methodology in the construction of the structural interactome in this manuscript compared to the previous one? If yes, that aspect should be described. If no, I would suggest just citing their study and stating that the structural interactome in Ghadie and Xua 2019, Nat Comm. was used. I will have additional questions about the structural interactome in the other sections.

Results

4. The first two sections of the Results are a repetition of Ghadie and Xua 2019, Nat Comm. maybe with some dataset updates. My question is why the authors did not use the already available dataset that has been curated in their previous publication directly and label the transient and permanent PPIs but started the same pipeline from scratch and perform the same analysis to construct structural interactome, interface extraction and mutation mapping again. The data is already there. I understand that there might be some updates in PPIs and datasets but still it will be an addition to the already available one.

5. The work classifies the permanent and transient interactions based on the calculated dG values of FoldX. How good is FoldX in predicting transient and permanent interactions? Finding a ground truth dataset having already known permanent and transient interactions and calculating the precision, recall and other performance measures would support that the labelled permanent and transient interactions are at some accuracy or precise.

6. My previous comment is also valid for the overall pipeline. The authors have some assumptions based on tissue- or temporal- expression to classify transient and permanent interactions. However, there is no validation on a set of already known transient and permanent interactions. Applying their classification approach to a benchmark dataset and evaluate their performance would make a solid basis for their classification. A relevant paper to find a benchmark dataset might be PMC4084939. There are more articles I presume cataloging a benchmark dataset.

7. Authors collect the expression levels from GEO times series datasets and Illumina Body Map tissue expression dataset. More information needs to be added about these datasets. Are the samples from healthy people or from a patient? Are there the mutation profiles available for these samples in the same data sources? If yes, how much overlap is present between the sample mutation profiles and the mutations collected from dbSNP and ClinVar? The follow-up question would be how this information would affect their analysis, if available?

8. I think the novelty aspect of the study is hidden in the expression-based classification of permanent and transient interactions. However, interestingly, this aspect is not presented in the current manuscript as much as it deserves. Actually, Table 1 summarizes their overall results, but it is rather orphan in the text and not explained in detail in the Result section to give a detailed understanding about the results.

Other Comments

9. Discussion section reads like an extended abstract of the manuscript. It needs a detailed rewriting and discussing the results based on what available literature.

10. I strongly encourage the authors depositing all relevant codes to a repository and making available for the reproducibility of the study.

11. Author’s summary section repeats the Abstract. It is expected to provide a non-technical summary of the work. I suggest reconsidering this aspect and revising the summary.

**Have the authors made all data and (if applicable) computational code underlying the findings in their manuscript fully available?**

Reviewer #1: Yes

Reviewer #2: None

Reviewer #3: **No: **The source codes are not provided.

PLOS authors have the option to publish the peer review history of their article (what does this mean?). If published, this will include your full peer review and any attached files.

Reviewer #1: No

Reviewer #2: No

Reviewer #3: No
---

## [Decision Letter · Decision Letter 1]

11 Mar 2022

Dear Dr. Xia,

We are pleased to inform you that your manuscript 'Are transient protein-protein interactions more dispensable?' has been provisionally accepted for publication in PLOS Computational Biology.

Best regards,

Ozlem Keskin

Associate Editor

PLOS Computational Biology

Arne Elofsson

Deputy Editor

PLOS Computational Biology

Reviewer's Responses to Questions

**Comments to the Authors:**

Reviewer #1: Dear authors,

I would like to thank you for having taken all comments very seriously and addressing them carefully. I should also make amend for being a little rough on the data, forgetting too quickly how difficult experimental data are produced.

I have one last comment about the dispensable PPI based on the now clear definition, thank you. The dispensable PPI being those disturb by mutations at the interface and as a result disappearing from the interactomes with no impact on the fitness of the organism, could that define them as backup PPI (rather than dispensable in the sense of being useless for the fitness)? If their disappearance has no impact that could be because there is another PPI (or several other PPI) acting likewise. The sustainability of living systems is after all based on diversity of solutions, meaning backup /alternative ways. Maybe those dispensable PPI support the sustainability of the interactomes ie its future capacity to cope with perturbations, if removed the organism might become less and less sustainable, because it has a lower diversity.

Reviewer #2: THe authors addressed all of my comments.

**Have the authors made all data and (if applicable) computational code underlying the findings in their manuscript fully available?**

Reviewer #1: Yes

Reviewer #2: None

PLOS authors have the option to publish the peer review history of their article (what does this mean?). If published, this will include your full peer review and any attached files.

Reviewer #1: **Yes: **Claire Lesieur

Reviewer #2: No

---

## [Editor Report · Acceptance letter]

28 Mar 2022

PCOMPBIOL-D-21-01091R1 

Are transient protein-protein interactions more dispensable?

Dear Dr Xia,

I am pleased to inform you that your manuscript has been formally accepted for publication in PLOS Computational Biology. Your manuscript is now with our production department and you will be notified of the publication date in due course.

With kind regards,

Livia Horvath
